# Nociceptive sensory neurons promote CD8 T cell responses to HSV-1 infection

Jessica Filtjens[1,6], Anais Roger [1,6], Linda Quatrini [1,2], Elisabeth Wieduwild[1], Jordi Gouilly [1], Guillaume Hoeffel [1], Rafaëlle Rossignol[1], Clara Daher[1,3], Guilhaume Debroas[1], Sandrine Henri [1], Claerwen M. Jones [4], Bernard Malissen [1], Laura K. Mackay [4], Aziz Moqrich[5], Francis R. Carbone[4] & Sophie Ugolini [1✉]

Host protection against cutaneous herpes simplex virus 1 (HSV-1) infection relies on the induction of a robust adaptive immune response. Here, we show that $Nav_{1.8}^+$ sensory neurons, which are involved in pain perception, control the magnitude of CD8 T cell priming and expansion in HSV-1-infected mice. The ablation of $Nav_{1.8}$-expressing sensory neurons is associated with extensive skin lesions characterized by enhanced inflammatory cytokine and chemokine production. Mechanistically, $Nav_{1.8}^+$ sensory neurons are required for the downregulation of neutrophil infiltration in the skin after viral clearance to limit the severity of tissue damage and restore skin homeostasis, as well as for eliciting robust CD8 T cell priming in skin-draining lymph nodes by controlling dendritic cell responses. Collectively, our data reveal an important role for the sensory nervous system in regulating both innate and adaptive immune responses to viral infection, thereby opening up possibilities for new therapeutic strategies.

[1] Aix Marseille Univ, CNRS, INSERM, CIML, Centre d'Immunologie de Marseille-Luminy, Marseille, France. [2] Department of Immunology, IRCCS Bambino Gesù Children's Hospital, Rome, Italy. [3] Université de Paris, CNRS, Institut Cochin, INSERM, CNRS, Paris, France. [4] Department of Microbiology and Immunology, The University of Melbourne at The Peter Doherty Institute for Infection and Immunity, Melbourne, VIC, Australia. [5] Aix-Marseille-Université, CNRS, Institut de Biologie du Développement de, Marseille, France. [6] These authors contributed equally: Jessica Filtjens, Anais Roger. ✉email: ugolini@ciml.univ-mrs.fr

Host protection against infection is mainly mediated by the immune system. However, recent studies have revealed that the nervous system plays an important role in controlling inflammatory processes during infectious diseases. As a barrier organ, the skin is one of the first lines of defence against pathogens. At steady state, the skin-resident immune system includes macrophages, dendritic cells (DC), mast cells and T cell subsets[1]. These cells act as sentinels, reacting to physical injuries or pathogens by mounting a robust inflammatory response inducing the recruitment of additional immune cells. Neutrophils are among the first cells to migrate from the blood to the site of the assault. They can kill pathogens and produce cytokines, chemokines and proteolytic enzymes promoting the recruitment and activation of other cells, including monocytes[2]. Skin DC take up foreign antigens and present them to naive T cells in the skin-draining lymph nodes (dLN), initiating an adaptive T cell response[3,4]. The early events occurring at the site of infection are therefore crucial for the development of protective immunity.

The skin is also innervated by a dense meshwork of low- and high-threshold sensory nerves[5,6]. These nerves include the nociceptive sensory neurons (also called nociceptors), which specialise in detecting noxious stimuli and eliciting pain perception. During inflammatory processes and infections, pathogen-derived molecules, lipids and immune cell-derived mediators act on the peripheral nerve terminals of nociceptive sensory neurons[7]. These mediators include ATP, prostaglandins and leukotrienes, bradykinin, histamine, growth factors and cytokines[8]. These mediators bind to their receptors on the neurons, modifying neuron excitability and increasing action potential generation. The resulting signals are transduced to the spinal cord and relayed to the brain for processing, leading to the perception of pain. Neuronal responses to pathogens and injury are induced over a scale of milliseconds, whereas immune cell response induction takes hours or days. The early response of neurons may therefore make an important contribution to host defence, the elimination of threats and tissue repair processes. Upon activation, nociceptive sensory neurons release a number of mediators locally that can modulate the activity of immune cells present in the damaged skin[8]. However, depending on the experimental model considered, these neurons may have either pro- or anti-inflammatory activities, suggesting that their regulatory role depends on the pathological context[9–11].

Nociception has long been known to be modulated during infections caused by members of the herpesvirus family[12,13]. One of these viruses, herpes simplex virus type 1 (HSV-1), is a neurotropic virus highly prevalent in the human population. HSV-1 is transmitted principally by oral contact and causes infections in or around the mouth, which may manifest as painful blisters or ulcers. After the initial mucocutaneous disease, the virus infects the peripheral nervous system. Infected individuals often experience a tingling, itching or burning sensation around the mouth, before the appearance of skin/mucosal lesions, indicating an early activation of nociceptive pathways. However, the potential regulatory role of the activation of nociceptive sensory neurons on the immune response to HSV-1 has yet to be determined.

We addressed this question, using a well-characterised mouse model of cutaneous HSV-1 infection[14,15]. In this infectious model, HSV-1 goes through two phases of acute viral replication in the skin[14]. The first phase occurs in the keratinocytes surrounding the site of infection on the flank of the mouse. The virus then gains access to the peripheral terminals of sensory neurons, subsequently spreading by axonal transport to the sensory ganglia, where it undergoes extensive replication. The virions then travel by anterograde transport along the many axons derived from the infected ganglia, until they reach the skin, where

a secondary growth phase occurs, involving the entire dermatome innervated by the ganglia and giving rise to a zosteriform band of skin lesions. Our objective was to analyse the possible regulatory role of nociceptor sensory neurons in antiviral immunity by monitoring the early steps of innate and adaptive immune responses to HSV-1 in a genetic mouse model in which the large majority of nociceptors are ablated[16]. In these $Nav_{1.8}$-DTA mice, the diphtheria toxin (DTA) is used to ablate the sensory neurons expressing the sodium channel $Nav_{1.8}$. However, the absence of certain neuron subpopulations in $Nav_{1.8}$-DTA mice may, theoretically, have a direct effect on the rate of viral replication during the secondary growth phase, with possible consequences for the quality or magnitude of the immune response induced. We overcame this potential bias, which would otherwise have made it impossible to assess the direct role of neurons in controlling the immune response, by generating a mutant form of the virus capable of replicating in skin epithelial cells, but not in neurons.

Here, we show that nociceptive sensory neurons are required to limit inflammatory cytokine production in response to cutaneous HSV-1 infection, limiting neutrophil infiltration and promoting tissue repair. This regulation is necessary to elicit CD8 T cell priming by DC in skin-dLN. These findings reveal a novel aspect of neuroimmune interactions highlighting a functional role of the nervous system in promoting host antiviral immune response.

## Results

**Nociceptor-deficient mice display enhanced skin inflammation upon HSV-1 infection.** HSV-1 expresses a gene encoding a thymidine kinase enzyme essential for viral replication in neurons but not in epithelial cells[17]. Following infection with thymidine kinase (TK)-deficient forms of HSV-1, only the first phase of replication in the skin occurs[18]. We generated a virus with a mutation of the TK gene that expressed the SIINFEKL peptide derived from ovalbumin (OVA) (hereafter referred to as HSV-OVA-TK⁻), making it possible to monitor both the innate and adaptive antiviral responses following the primary phase of skin infection. We infected C57BL/6 mice with the HSV-OVA-TK⁻ or parental HSV-OVA virus. Viral titres in the skin two days post infection (pi) were similar (Fig. 1a), showing that the primary phase of viral replication was not affected by the deletion of the TK-encoding gene. We, therefore, used this HSV-OVA-TK⁻ viral mutant to analyse the role of $Nav_{1.8}^+$ sensory neurons during the early stage of cutaneous HSV-1 infection. Transcriptomic data available from public databases (http://www.immgen.org; http://biogps.org) show that the gene encoding $Nav_{1.8}$, *Scn10a*, is expressed in DRG neurons, but not in immune cells. For confirmation of this finding, we performed a fate mapping analysis in which we crossed $Nav_{1.8}^{Cre}$ mice with Rosa26-TdTomato mice. In this model, every $Nav_{1.8}$-expressing cell expresses the fluorescent marker TdTomato. Immunofluorescence analysis on the skin of these mice (Fig. 1b) clearly showed that the fluorescent marker was expressed in nerve fibres, but not in any other cell types, including CD45⁺ haematopoietic cells, in the skin. These data were confirmed by flow cytometry analysis, which also showed that CD45⁺ haematopoietic cells in the skin, lymph nodes (LN) and spleen did not express the fluorescent marker (Supplementary Fig. 1a). These data confirm the absence of $Nav_{1.8}$ expression in immune cells in this model, validating $Nav_{1.8}$-DTA mice as a relevant genetic model for studies of the possible regulatory role of $Nav_{1.8}$ in antiviral immune responses.

We, therefore, investigated the viral and immune processes occurring in the skin of nociceptor-deficient $Nav_{1.8}$-DTA and control DTA mice after HSV-1 infection. In control DTA mice, scarification of the upper dorsal flank and infection with HSV-OVA-TK⁻ resulted in minor skin lesions at the site of

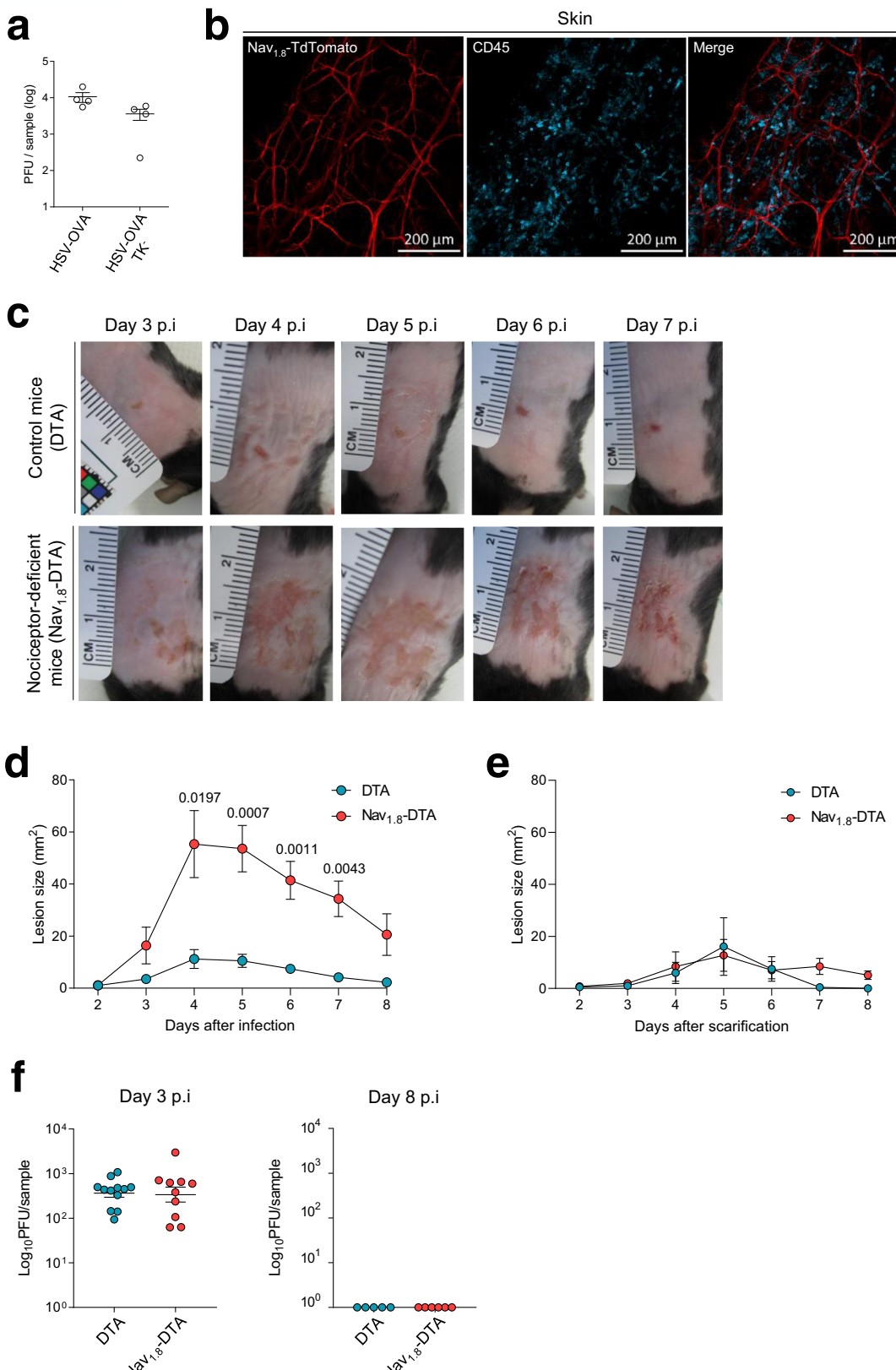

scarification, which gradually resolved (Fig. 1c and Supplementary Table 1). By contrast, infected $Nav_{1.8}$-DTA mice displayed much more extensive inflammatory lesions of the skin than their littermate controls, with these lesions appearing from day 3 pi and persisting beyond day 8 (Fig. 1c, d and Supplementary Table 1). Scarification alone, in the absence of infection, induced

minor lesions of similar size in $Nav_{1.8}$-DTA and control DTA mice (Fig. 1e and Supplementary Table 1).

We then examined whether the more severe lesions, observed in infected $Nav_{1.8}$-DTA mice, were due to higher viral loads in the skin of these mice. Three days after infection, viral titres were similar in infected DTA and $Nav_{1.8}$-DTA littermates, showing

**Fig. 1 Cutaneous HSV-1 infection in Nav$_{1.8}$-DTA mice.** Control DTA and Nav$_{1.8}$-DTA mice were infected with HSV-OVA-TK$^-$ after flank scarification. **a** C57BL/6 mice were infected with the HSV-OVA or HSV-OVA-TK$^-$ viral strains. Viral titres in the skin were analysed 2 days pi, $n = 4$ mice per group. **b** Immunofluorescence staining of Nav$_{1.8}$-TdTomato mouse skin. Whole-mount skin samples were stained with anti-CD45 antibodies (blue); Nav$_{1.8}^+$ cells express TdTomato (red), $n = 2$ samples. **c** Representative pictures of the skin lesion progression in control DTA (top line) and Nav$_{1.8}$-DTA mice (bottom line). **d** Size of the skin lesions of control DTA and Nav$_{1.8}$-DTA mice after HSV-OVA-TK$^-$ infection ($n = 19$–20 mice per group) or **e** treated with PBS only, after flank scarification ($n = 5$–9 mice per group) (the data obtained for each mouse are shown in Supplementary Table 1). $P$ values were obtained by mixed-effect analysis followed by Sidak's multiple comparison test. **f** Viral titres in the skin of control DTA and Nav$_{1.8}$-DTA mice after HSV-OVA-TK$^-$ infection on days 3 pi ($n = 13$ DTA mice and $n = 11$ Nav$_{1.8}$-DTA mice) and 8 pi ($n = 5$ DTA mice and $n = 6$ Nav$_{1.8}$-DTA mice). Data are presented as mean ± SEM.

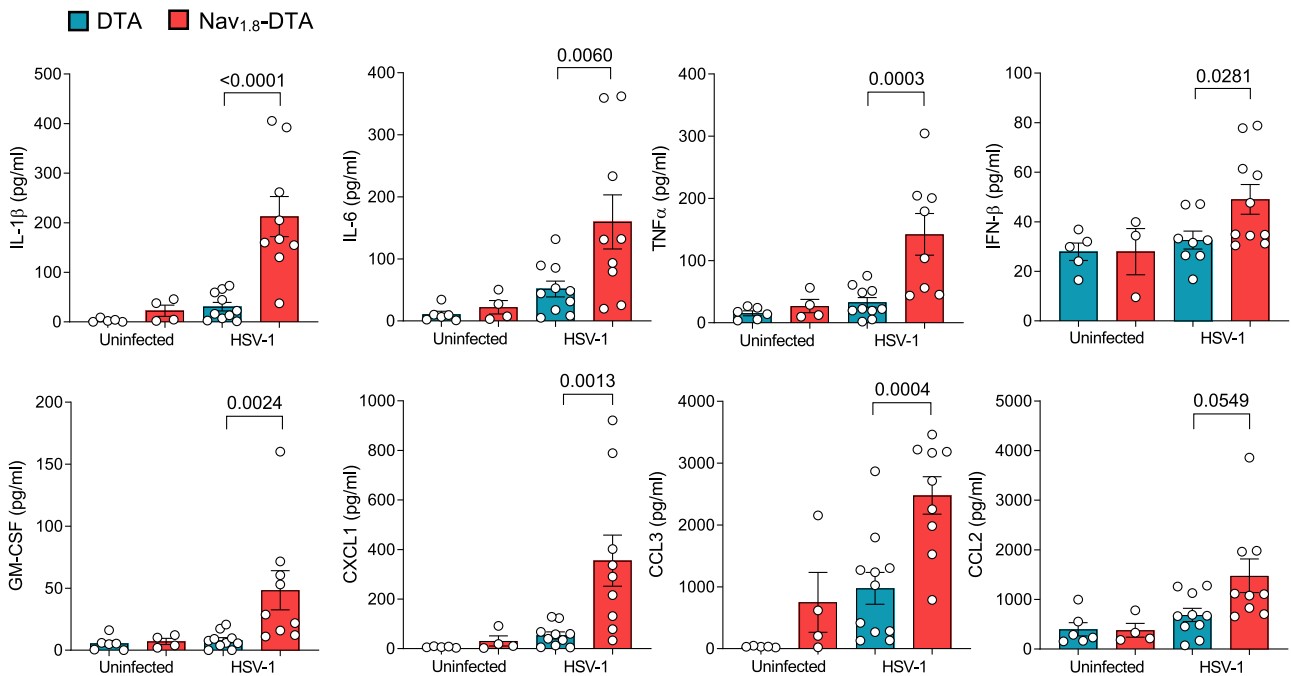

**Fig. 2 Excessive skin inflammation in Nav$_{1.8}$-DTA mice following cutaneous HSV-1 infection.** Cytokine and chemokine concentrations in skin homogenates of control DTA (blue bars) and Nav$_{1.8}$-DTA (red bars) mice 6 days after HSV-OVA-TK$^-$ infection or in PBS-treated mice. Data from three independent experiments are presented as mean ± SEM; $P$ values were obtained by using one-way ANOVA followed by Sidak's multiple comparison test. Each dot represents the data obtained for one mouse. $N = 4$–10 mice per group.

that the lack of nociceptive sensory fibres in Nav$_{1.8}$-DTA mice did not affect the course of HSV-OVA-TK$^-$ viral replication (Fig. 1f). In addition, 8 days pi, the virus had been cleared in both control DTA and Nav$_{1.8}$-DTA littermates (Fig. 1f). These data are consistent with replication of the HSV-OVA-TK$^-$ virus occurring upstream from the ganglia and being unaffected by the loss of Nav$_{1.8}^+$ sensory neurons. In agreement with these data, we observed no difference in body weight between control DTA and Nav$_{1.8}$-DTA mice during infection (Supplementary Fig. 1b). Therefore, viral replication is similar in control DTA and Nav$_{1.8}$-DTA mice and the skin lesions observed in mice lacking nociceptive sensory neurons are not due to a defect in viral clearance.

We then investigated whether the larger lesions seen in nociceptor-deficient mice resulted from dysregulation of local inflammatory processes. HSV-1 infection induces the expression of inflammatory chemokines and cytokines, including CCL2, CCL3, CXCL1, IL-1β, IL-6, TNF-α and type I IFNs[19–21]. We analysed the effect of nociceptor loss on inflammatory responses, by measuring cytokine and chemokine levels in the skin of Nav$_{1.8}$-DTA and control DTA mice. Six days pi with the HSV-OVA-TK$^-$ strain, levels of IL-1β, IL-6, TNFα, IFN-β and GM-CSF in the skin were higher in infected Nav$_{1.8}$-DTA mice than in their control littermates

(Fig. 2). Furthermore, the chemokines CXCL1 (KC), CCL3 (MIP-1α) and CCL2 (MCP-1) were present in larger amounts in skin samples from infected Nav$_{1.8}$-DTA mice than in skin samples from DTA controls. By contrast, the production of IL-17 and IL-23 was not induced in the skin of these mice upon scarification or HSV-1 infection. The concentrations of these two cytokines in the skin were below the detection threshold of the assay used (10 pg/ml). Collectively, these data demonstrate that nociceptive sensory neurons are required to limit the magnitude of the inflammatory response induced in the skin upon HSV-1 infection.

**Nociceptive sensory neurons regulate monocyte activation and neutrophil infiltration in HSV-1-infected skin.** Following infection, despite normal HSV-1 clearance, higher levels of inflammatory cytokines and chemokines were detected in the skin of nociceptor-deficient mice. These mediators are involved in the activation of tissue-resident cells and in the recruitment of neutrophils and monocytes to the infection site[22]. We, therefore, analysed the skin immune response in nociceptor-deficient and -sufficient mice on day 6 pi. Skin cells were isolated from infected Nav$_{1.8}$-DTA and DTA mice and analysed, by flow cytometry, for their expression of the markers Ly6G, CD11c, CD11b, CD206, MHC-II, CD24, c-kit, CD4 and TCR γδ (Fig. 3a, b). The

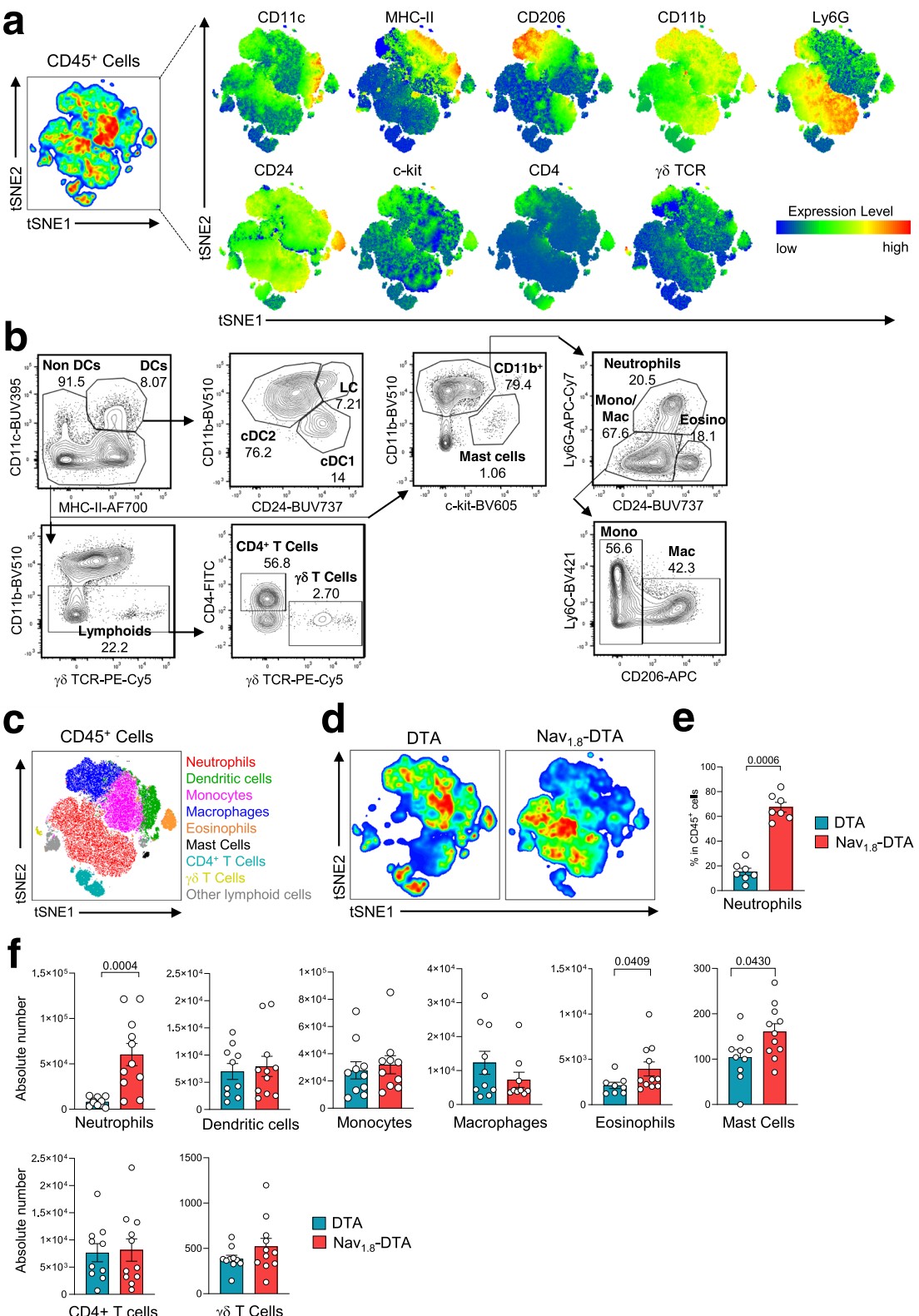

distributions and relative frequencies of CD45$^+$ immune cell subsets were then compared in an unsupervised t-distributed stochastic neighbour embedding (tSNE) analysis (Fig. 3c, d). Using this clustering analysis, we were able to distinguish several subsets that phenotypically correspond to neutrophils (CD11b$^+$ Ly6G$^+$), eosinophils (CD11b$^+$ Ly6G$^-$, CD24$^{high}$), dendritic cells (DC, CD11c$^{high}$ MHC-II$^+$), monocytes (CD11b$^+$ Ly6G$^-$ CD24$^-$ CD11c$^-$CD206$^-$Ly6C$^{high/low}$), macrophages (CD11b$^+$ Ly6G$^-$

CD24$^-$ CD11c$^-$ CD206$^+$ CD64$^+$) and T cell subsets (Fig. 3a, b). The relative clustering of the various individual control DTA and Nav$_{1.8}$-DTA mice used for the tSNE analysis revealed a large increase in the frequency of neutrophils in the skin of nociceptor-deficient mice (Fig. 3d, e). Neutrophil counts were also higher in the skin of Nav$_{1.8}$-DTA mice, whereas the numbers of monocytes, macrophages, DC subsets, δγ T cells, eosinophils, CD4 T cells and mast cells were unchanged or only slightly different from those in

**Fig. 3 Nociceptor-deficient mice present higher levels of neutrophil infiltration in the skin after HSV-1 infection.** Control DTA and $Nav_{1.8}$-DTA mice were infected with HSV-OVA-TK$^-$ by flank scarification. Immune responses in the skin were analysed 6 days pi by flow cytometry. **a** tSNE analysis of CD45$^+$ cells with the CD11c, MHC-II, CD206, CD11b, Ly6G, CD24, c-kit, CD4 and γδ TCR markers. The colours in the expression-level heatmaps (right panel) represent the median intensity values for each marker. **b** Flow cytometry gating strategy used to identify immune cell subsets: the CD45$^+$ cell population was selected after the exclusion of doublets and dead cells. DCs were gated as CD11c$^{high}$ MHC-II$^+$ cells and split into subsets on the basis of their expression of CD24 and CD11b. cDC1 was defined as CD24$^+$ CD11b$^-$ DC, cDC2 as CD11b$^+$ CD24$^-$ DC and Langerhans cells (LC) as CD24$^+$ CD11b$^+$ DC. Neutrophils were gated as CD11b$^+$ Ly6G$^+$ cells. Eosinophils were then gated as CD11b$^+$ CD24$^+$ cells. Monocytes and macrophages were defined as CD11b$^+$ CD24$^-$ cells and subdivided into macrophages (CD11b$^+$ CD24$^-$ CD206$^+$ CD64$^+$) and monocytes (CD11b$^+$ CD24$^-$ CD206$^-$ Ly6C$^{high/low}$). Finally, within the non-DC subset, lymphoid cells were gated as CD11b$^-$ γδ TCR$^+$ cells and subdivided into CD4$^+$ T cells (CD4$^+$ γδ TCR$^-$), γδ T cells (CD4$^-$ γδ TCR$^+$) and other lymphoid cells (CD4$^-$ γδ TCR$^-$). **c** Distribution of the different immune cell subsets in the tSNE analysis: neutrophils (CD11b$^+$ Ly6G$^+$, in red), dendritic cells (CD11c$^{high}$ MHC-II$^+$, in green), monocytes (CD11b$^+$ Ly6G$^-$ CD24$^-$ CD206$^-$, in pink), macrophages (CD11b$^+$ Ly6G$^-$ CD24$^-$ CD206$^+$, in blue), eosinophils (CD11b$^+$ Ly6G$^-$ CD24+, in orange), mast cells (CD11b$^+$ c-kit$^+$, in black), CD4$^+$ T cells (CD11b$^-$ TCRγδ$^-$ CD4$^+$, in cyan), γδ T cells (CD11b$^-$ TCRγδ$^+$, in dark yellow) and other lymphoid cells (CD11b$^-$ TCRγδ$^-$ CD4$^-$, in grey). **d** Relative clustering of skin immune cells from the indicated mouse strains (control DTA or $Nav_{1.8}$-DTA) used for the tSNE analysis. **e** Percentages of neutrophils among CD45$^+$ skin cells from DTA and $Nav_{1.8}$-DTA mice 6 days pi. **f** Absolute numbers of immune cells subsets in the skin of DTA and $Nav_{1.8}$-DTA mice 6 days pi. The data presented are from three independent experiments ($n = 8$–11 mice per group) and are shown as mean ± SEM; P values were obtained by using a Mann–Whitney test (two-tailed).

control mice (Fig. 3f). $Nav_{1.8}^+$ sensory neurons are thus required to limit neutrophil infiltration in the skin after HSV-1 infection.

We then analysed the cellular source of the inflammatory cytokines produced in excess in the skin of $Nav_{1.8}$-DTA mice. Intracellular staining and flow cytometry analysis of skin cells on day 6 pi revealed higher levels of TNF-α and IL-1β production by monocytes from $Nav_{1.8}$-DTA mice (Fig. 4a, c). The percentages of neutrophils producing inflammatory cytokines were similar in DTA and $Nav_{1.8}$-DTA mice, showing that sensory neurons are not required to regulate the activation of these cells (Fig. 4b). However, because the absolute number of neutrophils increased (Fig. 3f), the numbers of TNF-α-, IL-6- and IL-1β-producing neutrophils were higher in nociceptor-deficient mice than in their control littermates (Fig. 4d). By contrast, for the other immune cell subsets, the absolute numbers of cytokine-producing cells were similar in the two genotypes (Supplementary Fig. 2a–c).

These data show that nociceptive sensory neurons down-regulate inflammatory cytokine production by monocytes and control neutrophil influx into the skin following HSV-1 infection. This regulation is required to limit the levels of TNF-α, IL-1β and IL-6 in the skin after infection.

**The CD8 T cell response to HSV-1 is impaired in nociceptor-deficient mice.** In the epidermis of control mice, we observed that Langerhans cells (LC) were in close contact with PGP9.5+ nerve terminals, which were absent in the skin of $Nav_{1.8}$-DTA mice (Fig. 5a). We, therefore, analysed whether sensory neurons could also contribute to regulate the adaptive immune response to HSV-1. We first monitored the DC response in more detail by analysing the cDC1 (CD11c$^+$MHC-II$^+$ CD24$^+$CD11b$^-$), cDC2 (CD11c$^+$MHC-II$^+$ CD24$^-$ CD11b$^+$) and LC (CD11c$^+$MHC-II$^+$ CD24$^+$CD11b$^+$) subsets (Fig. 3b). The numbers of DC subtypes (cDC1, cDC2, LC) were similar in the skin of uninfected $Nav_{1.8}$-DTA and control DTA mice (Supplementary Fig. 3a–c, left panels). We found no difference in DC counts on days 2 and 4 pi (Supplementary Fig. 3a, b, right panels). By contrast, 6 days pi, the numbers of cDC1 and LC were lower in the skin of $Nav_{1.8}$-DTA mice (Fig. 5b), suggesting that the DC response is affected in the absence of nociceptive neurons. Upon HSV-1 infection, skin DC migrate from the skin to the draining lymph nodes, where they cross-present viral antigens to CD8 T cells[18,23–25]. Previous studies have shown that DC migrating from the skin can transfer antigens to LN-resident DC, facilitating T cell activation[24,25]. We found no difference in the frequency of DC subsets in the skin-draining LN of infected $Nav_{1.8}$-DTA and control DTA mice (Supplementary Fig. 4a–d). However, it is possible that regulation

of the DC response in the skin of nociceptor-deficient mice affects the quality of T cell priming in the dLN. We, therefore, investigated whether purified DC isolated from the skin-draining LN of DTA and $Nav_{1.8}$-DTA mice were able to prime naive CD8 T cells in vitro. $Nav_{1.8}$-DTA and DTA mice were infected with the HSV-OVA-TK$^+$ virus. On day 4 pi, CD11c$^+$ DC were isolated from brachial and axillary LN and used to activate fluorescently (cell trace violet)-labelled OVA-specific CD8 T cells isolated from OT-I transgenic mice. These T cells (hereafter referred to as OT-I T cells) carried a Vα2/Vβ5 T cell receptor (TCR) specific for the OVA peptide SIINFEKL presented in the context of major histocompatibility complex (MHC)-class I H2-K$^b$ [26]. CD8 T cell proliferation was analysed by measuring the decrease in the mean fluorescence intensity (MFI) of the labelled T cells on day 4 of coculture. Following the addition of exogenous OVA peptide to the cell culture, OT-I T cells proliferated similarly whether they were incubated with DC from $Nav_{1.8}$-DTA or control DTA mice (Fig. 5c). By contrast, in the absence of exogenous peptide, DC from $Nav_{1.8}$-DTA-infected mice were unable to induce the proliferation of OT-I T cells to levels similar to those observed with control DC (Fig. 5d). These data suggest that the ability of DCs to present viral antigens in the dLN is impaired in mice lacking $Nav_{1.8}^+$ sensory neurons.

We therefore further analysed whether nociceptive neurons were required to induce a robust CD8 T cell response in vivo. We infected $Nav_{1.8}$-DTA or control DTA littermate mice with the HSV-OVA-TK$^-$ virus. On the day before infection, we performed an adoptive transfer procedure, in which $Nav_{1.8}$-DTA and DTA mice received $5 \times 10^4$ congenic CD45.1$^+$ OT-I T cells (Fig. 5e). We analysed the virus-specific T cell response in the skin-draining lymph nodes (dLN) of DTA and $Nav_{1.8}$-DTA littermates 4 and 8 days pi, by monitoring the expansion of the transferred OT-I T cells. We observed an expansion of the OT-I T cell population 8 days pi, in the skin-draining LN of both mouse strains (Fig. 5f). However, OT-I T cell numbers were much lower in $Nav_{1.8}$-DTA mice than in their control DTA littermates, demonstrating that nociceptive sensory neurons are required to promote robust CD8 T cell expansion (Fig. 5f). Similarly, the numbers of OT-I T cells in the spleen of $Nav_{1.8}$-DTA mice were smaller than those in their DTA control littermates, indicating that this regulation can also affect the systemic immune response (Fig. 5g). By contrast, we observed no difference in skin T cell counts between infected nociceptor-deficient and -sufficient mice 8 days pi (Supplementary Fig. 4f), suggesting that nociceptive sensory neurons are not required to induce the recirculation of CD8 T cells from the periphery to the skin.

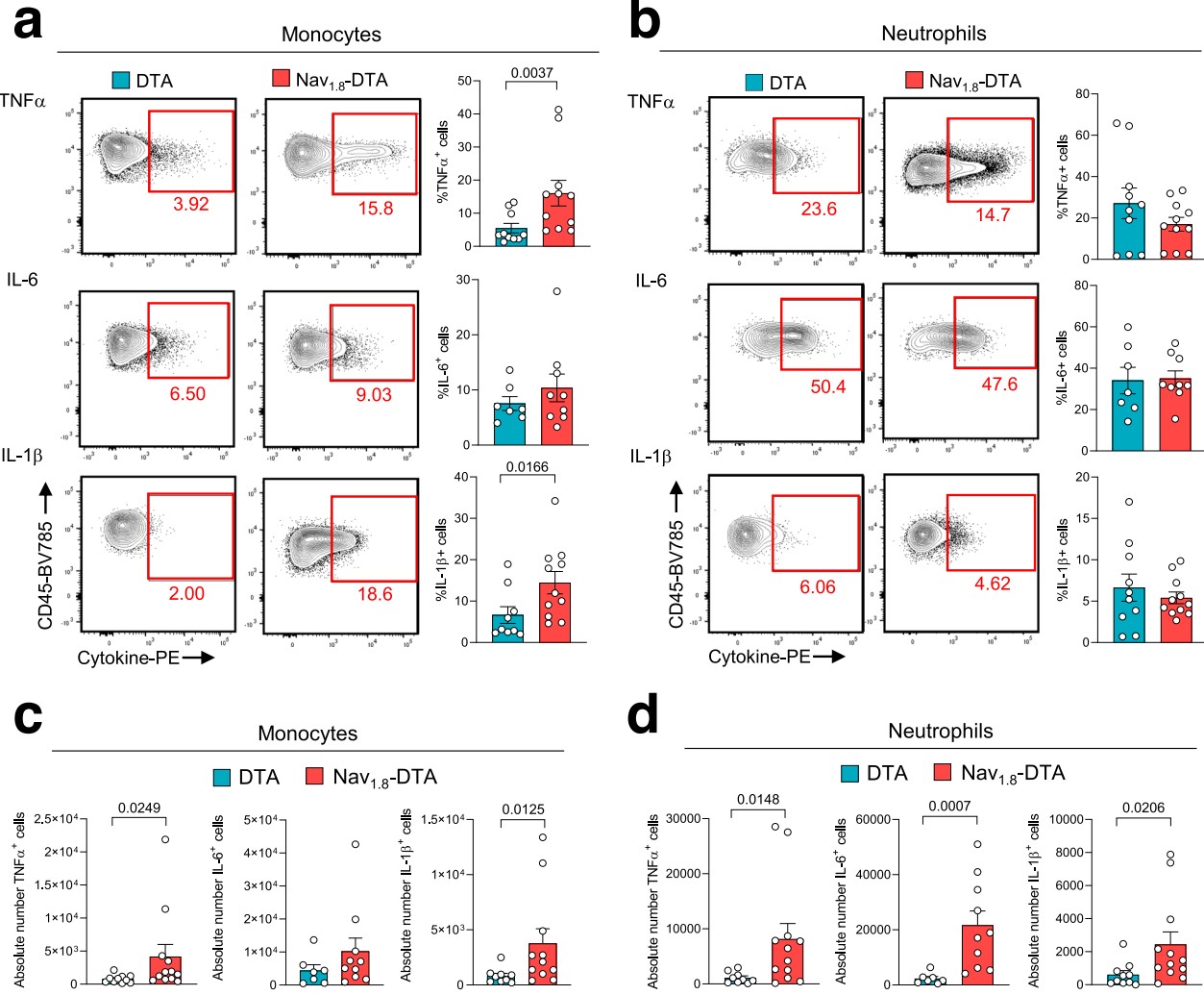

**Fig. 4 Cellular source of inflammatory cytokines.** Control DTA and Nav$_{1.8}$-DTA mice were infected with HSV-OVA-TK$^-$ by flank scarification. Inflammatory cytokine production by immune cell subsets in the skin was analysed by intracellular staining and flow cytometry analysis 6 days pi. **a**, **b** Representative dot plots and quantification of the % of TNF-α$^+$, IL-6$^+$ and IL-1b$^+$ monocytes (**a**) and neutrophils (**b**) in the skin of DTA and Nav$_{1.8}$-DTA mice 6 days pi. **c**, **d** Absolute numbers of TNF-α$^+$, IL-6$^+$ and IL-1β$^+$ monocytes (**c**) and neutrophils (**d**) in the skin of DTA and Nav$_{1.8}$-DTA mice 6 days pi. The results shown are from three independent experiments (*n* = 7–11 mice per group). The data are presented as mean ± SEM. *P* values were obtained by using a Mann–Whitney test (two-tailed).

**Nociceptor sensory neurons limit the severity of HSV-1-induced skin lesions by downregulating neutrophil recruitment in the skin.** Neutrophil recruitment in tissues is crucial, providing the first line of defence against invading pathogens. However, the neutrophil response must be tightly regulated to avoid excessive tissue damage and to restore homoeostasis[2]. With the aim of improving our understanding of the role of nociceptors in controlling the neutrophil response, we analysed the neutrophil influx in the skin of Nav$_{1.8}$-DTA and control DTA mice over the course of HSV-1 infection. Neutrophil counts had increased to a similar extent at day 4 pi, in both mouse strains, but they strongly decreased by 6 days pi, exclusively in DTA mice (Fig. 6a). By contrast, in Nav$_{1.8}$-DTA mice, neutrophil counts remained high at days 6 and 8 pi. Therefore, in the absence of nociceptive fibres, a high level of neutrophil infiltration persists after viral clearance. This effect was not observed for monocytes and macrophages, the distribution of which did not differ between control DTA and Nav$_{1.8}$-DTA mice (Supplementary Fig. 5a, b). However, the frequencies of monocytes producing TNF-α and IL-1β had increased by day 6 pi (Fig. 4c). It is thus possible that, after peak viral

replication, nociceptive sensory neurons downregulate monocyte activation and the production of inflammatory mediators by these cells, thereby decreasing neutrophil infiltration. These data reveal an important regulatory role for nociceptive sensory neurons in modulating the cutaneous innate immune response and, in particular, neutrophil infiltration in the skin during the phase of tissue repair initiated after viral clearance.

We further analysed the mechanisms by which neutrophil infiltration persisted on day 6 pi in Nav$_{1.8}$-DTA mice. The larger number of neutrophils may reflect an increase in recruitment, a defect in cell death or a higher level of retention in the tissue. A key feature of sites of inflammation is their low oxygen (O2) concentration (hypoxia). Hypoxia can influence the progression of inflammatory diseases and can inhibit neutrophil cell death[27]. The hypoxia-induced inhibition of neutrophil apoptosis is dependent on the expression of the hypoxia-inducible factor 1-alpha (HIF-1α) transcription factor[28]. HIF-1α is also required for dendritic cell maturation and the activation of T cells[29]. However, we found no difference in HIF-1α levels in the skin between Nav$_{1.8}$-DTA and control DTA mice (Fig. 6b). Consistently, the

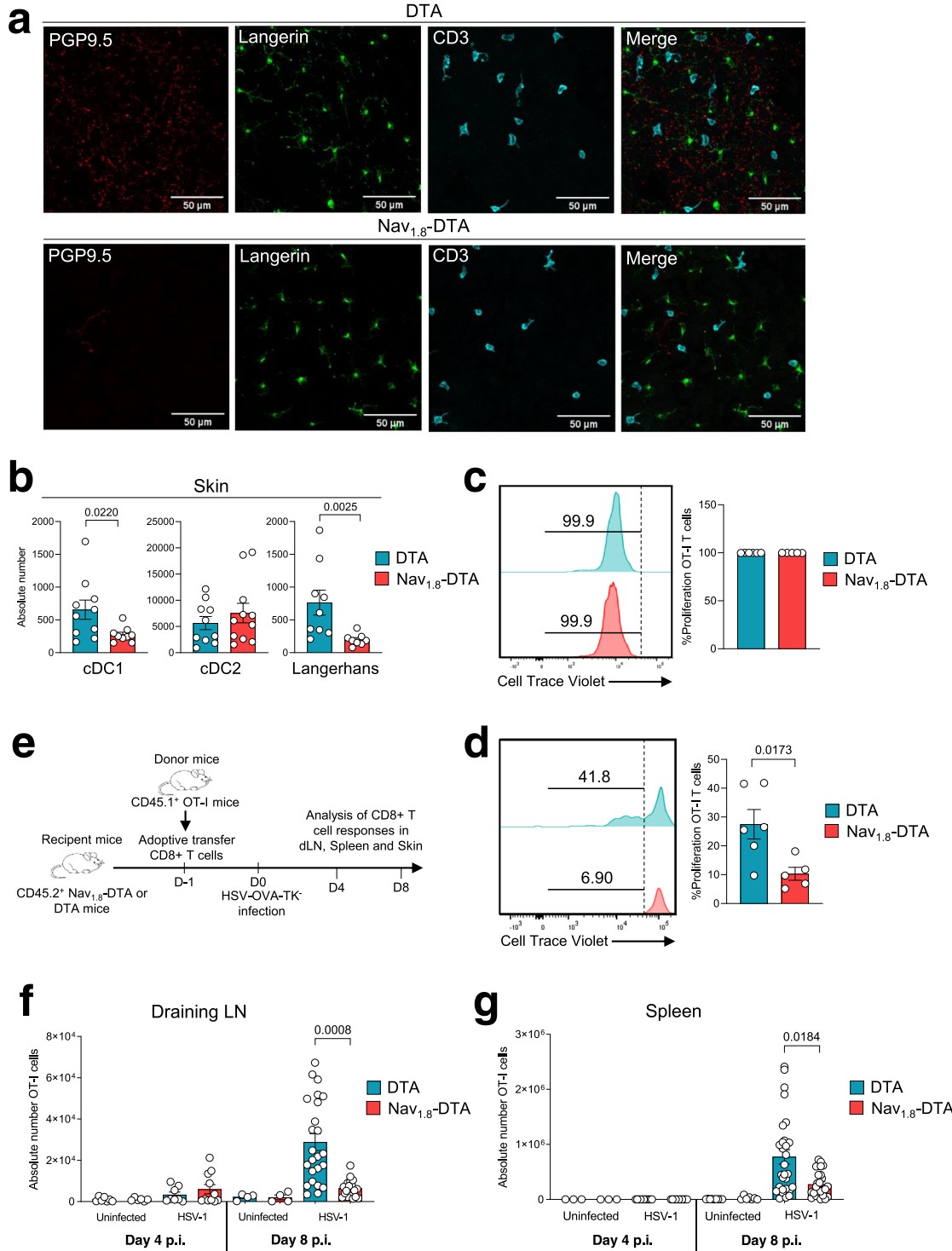

frequency of neutrophils undergoing cell death in the skin was also similar between mice of these two genotypes (Fig. 6c). These data suggest that the recruitment of neutrophils, rather than their survival, increases in the absence of nociceptive sensory neurons, consistent with the higher level of CXCL1 in the skin of Nav$_{1.8}$-DTA mice (Fig. 2).

The persistence of neutrophil recruitment observed in the absence of sensory neurons was associated with the induction of extensive tissue lesions at the site of infection (Fig. 1c, d). We, therefore, explored whether neutrophils were directly responsible

for the extensive cutaneous lesions observed in Nav$_{1.8}$-DTA mice. We depleted neutrophils by treating Nav$_{1.8}$-DTA and DTA mice with an anti-Ly6G-depleting antibody or an isotype control (IC) antibody 2 days after HSV-1 infection (Fig. 6d). Neutrophil depletion was sufficient to limit the size of skin lesions in Nav$_{1.8}$-DTA mice to that observed in control mice (Fig. 6e). These data suggest that, upon HSV-1 infection, the nociceptive sensory neurons prevent excessive skin inflammation and its associated damage mostly through the downregulation of neutrophil infiltration.

**Fig. 5 Nociceptors are required to promote a robust CD8 T cell response to HSV-1 infection. a** Immunofluorescence staining of the epidermis of DTA (top panels) and $Nav_{1.8}$-DTA (bottom panels) mice. Neurons were stained with anti-PGP9.5 antibodies (in red), Langerhans cells (LC) were stained with anti-langerin antibodies (in green) and T cells were stained with anti-CD3 antibodies (in blue). $N = 2$–3 mice per group. **b** Control DTA and $Nav_{1.8}$-DTA mice were infected with HSV-OVA-TK$^-$ by flank scarification. Skin DC subsets were analysed by flow cytometry using the gating strategy presented in Fig. 3b. Absolute numbers of cDC1, cDC2 and LC on day 6 pi are shown. Each dot represents the data obtained for one mouse. $N = 8$–11 mice per group. The data are presented as mean ± SEM. $P$ values were obtained by using a Mann–Whitney test (two-tailed). **c, d** In vitro activation of OT-I T cells. OT-I T cells from naive mice were purified, stained with a fluorescent marker and co-cultured for 48 h with purified CD11c$^+$ DCs from infected control DTA (blue) or $Nav_{1.8}$-DTA (red) brachial and axillary lymph nodes isolated on 4 days pi. Representative dot plots and quantification of the frequency of proliferating OT-I T cells in the presence (**c**) or absence (**d**) of the OVA peptide (SIINFEKL). $N = 5$–6 per group. The data are presented as mean ± SEM. $P$ values were obtained by using a Mann–Whitney test (two-tailed). **e** Experimental design used for monitoring antiviral CD8 T cell responses in vivo. Control DTA (blue) and $Nav_{1.8}$-DTA (red) recipient mice were injected with $5 \times 10^4$ naive virus-specific T cells (CD45.1$^+$ OT-I T cells) 1 day before HSV-OVA-TK$^-$ infection by flank scarification and were analysed 4 or 8 days pi. **f, g** Absolute numbers of virus-specific T cells (OT-I) in dLN ($n = 4$–24 mice per group) (**f**) and spleens ($n = 3$–30 per group) (**g**) were determined at the indicated time points. Virus-specific T cells were detected by flow cytometry and gated based on their expression of the Vα2, CD45.1, CD3 and CD8 markers, the gating strategy used is shown in Supplementary Fig. 4e. The results shown are representative of at least three independent experiments. The data are presented as mean ± SEM; $P$ values were obtained by using a Mann–Whitney test (two-tailed).

**Nociceptive sensory neuron-mediated control of neutrophil infiltration in the skin is required for induction of the antiviral CD8 T cell response.** Recent studies have suggested that neutrophils may play an important role in orchestrating adaptive immune responses[30,31]. In particular, neutrophils have been shown to inhibit T cell proliferation and promote T cell apoptosis in some models[31,32]. Neutrophils can also affect antigen presentation and DC migration to the LN[33–35]. We, therefore, investigated whether the larger numbers of neutrophils in the skin of nociceptor-deficient mice were responsible for the defective CD8 T cell response (Fig. 5f, g), by analysing the DC and T cell responses to HSV-1 infection in the skin of $Nav_{1.8}$-DTA and DTA mice treated with the anti-Ly6G antibody or with the IC on day 8 pi (Fig. 6f, g). Consistent with the lower frequency of cDC1 in the skin of $Nav_{1.8}$-DTA mice 6 days pi (Fig. 5b), we also observed significant fewer cDC1 in the skin of nociceptor-deficient mice than in littermate controls upon IC antibody treatment 8 days pi (Fig. 6f). This defect was reversed by neutrophil depletion with the anti-Ly6G antibody, suggesting that the downregulation of neutrophil recruitment by nociceptive sensory neurons could affect the cDC1 response in the skin. By contrast, no effect on the numbers of monocytes, macrophages and eosinophils was observed in the skin upon neutrophil depletion (Supplementary Fig. 5d).

We then investigated whether the impairment of the CD8 T cell response in mice lacking nociceptive sensory neurons was linked to the dysregulation of the neutrophil response in these mice. Consistent with our previous data (Fig. 5f, g), in mice treated with the IC antibody, the OT-I T cell population expanded less in the dLN and spleen of infected $Nav_{1.8}$-DTA mice than in those of DTA controls (Fig. 6g). By contrast, neutrophil depletion rescued this phenotype and restored normal T cell responses in both the spleen and the draining LN of $Nav_{1.8}$-DTA mice (Fig. 6g). Finally, consistent with the data presented in Supplementary Fig. 4f, no differences were observed in the cutaneous T cell response between $Nav_{1.8}$-DTA and control DTA mice (Supplementary Fig. 5e).

In conclusion, the control of neutrophil infiltration by sensory nerves is essential to elicit a robust adaptive CD8 T cell response. In the absence of this neuroimmune regulation, the DC response is impaired and virus-specific T cells are less efficiently primed in the dLN. These data demonstrate that sensory nociceptive neurons play an essential role in promoting a robust adaptive immune response to HSV-1 infection by controlling the extent of the inflammatory response in the skin.

**Nociceptors are not required for the control of T cell and neutrophil responses in a model of cutaneous vaccination.** We then investigated whether this regulation of innate and adaptive immune responses by nociceptive sensory neurons could also be observed in another model of skin inflammation.

Neutrophil and anti-OVA CD8 T cell responses were analysed in a skin vaccination model. We immunised mice with ovalbumin in complete Freund's adjuvant (CFA), which is commonly used to mimic microbial infections, skin inflammation and to induce a strong immune response[36]. We used this model because previous studies have shown that the hyperalgesia associated with CFA injections is lost in $Nav_{1.8}$-DTA mice[16], suggesting that $Nav_{1.8}$-expressing neurons are activated and could therefore play a role after the induction of inflammation by CFA.

The neutrophil influx was similar in the skin of $Nav_{1.8}$-DTA and DTA mice after intraplantar CFA injection in the hind paw of the mice (Fig. 7a). Moreover, the degree of skin inflammation, analysed by measuring changes in the thickness of the paw of the mouse, was similar for both genotypes (Fig. 7b). Finally, we found no difference in the frequency or the absolute number of OVA-specific CD8 T cells in the dLN of $Nav_{1.8}$-DTA and DTA mice upon immunisation (Fig. 7c, d). these data show that the regulation of the neutrophil and T cell responses by sensory neurons is context-dependent. Moreover, they also demonstrate that the neutrophil response is not intrinsically affected in $Nav_{1.8}$-DTA mice, highlighting the relevance of this genetic model.

## Discussion

Sensory neurons play multifaceted roles in maintaining homeostasis, detecting danger and eliciting protective immune responses. Recent studies have highlighted functional crosstalk between the nervous and immune systems during bacterial and fungal infections of the skin[8,9,11,37], but less is known about the role of neuroimmune interactions during host responses to viruses. Here, we used a genetic mouse model in which the ablation of $Nav_{1.8}$-expressing neurons results in an absence of inflammatory pain perception[16]. These mice were infected with a genetically modified form of HSV-1, able to replicate only in keratinocytes but not in neurons, which made it possible to analyse early neuroimmune interactions in the skin and their contribution to the host response.

Our findings reveal a previously unidentified role of nociceptive sensory neurons in controlling both the innate and adaptive immune responses to cutaneous HSV-1 infection.

Following cutaneous infection, nociceptor-deficient mice presented larger primary lesions at the infection site than their control littermates. However, viral clearance from the skin was similar in control DTA mice and $Nav_{1.8}$-DTA mice, excluding the possibility of the larger lesions in nociceptor-deficient mice being due to uncontrolled viral replication. This phenotype was

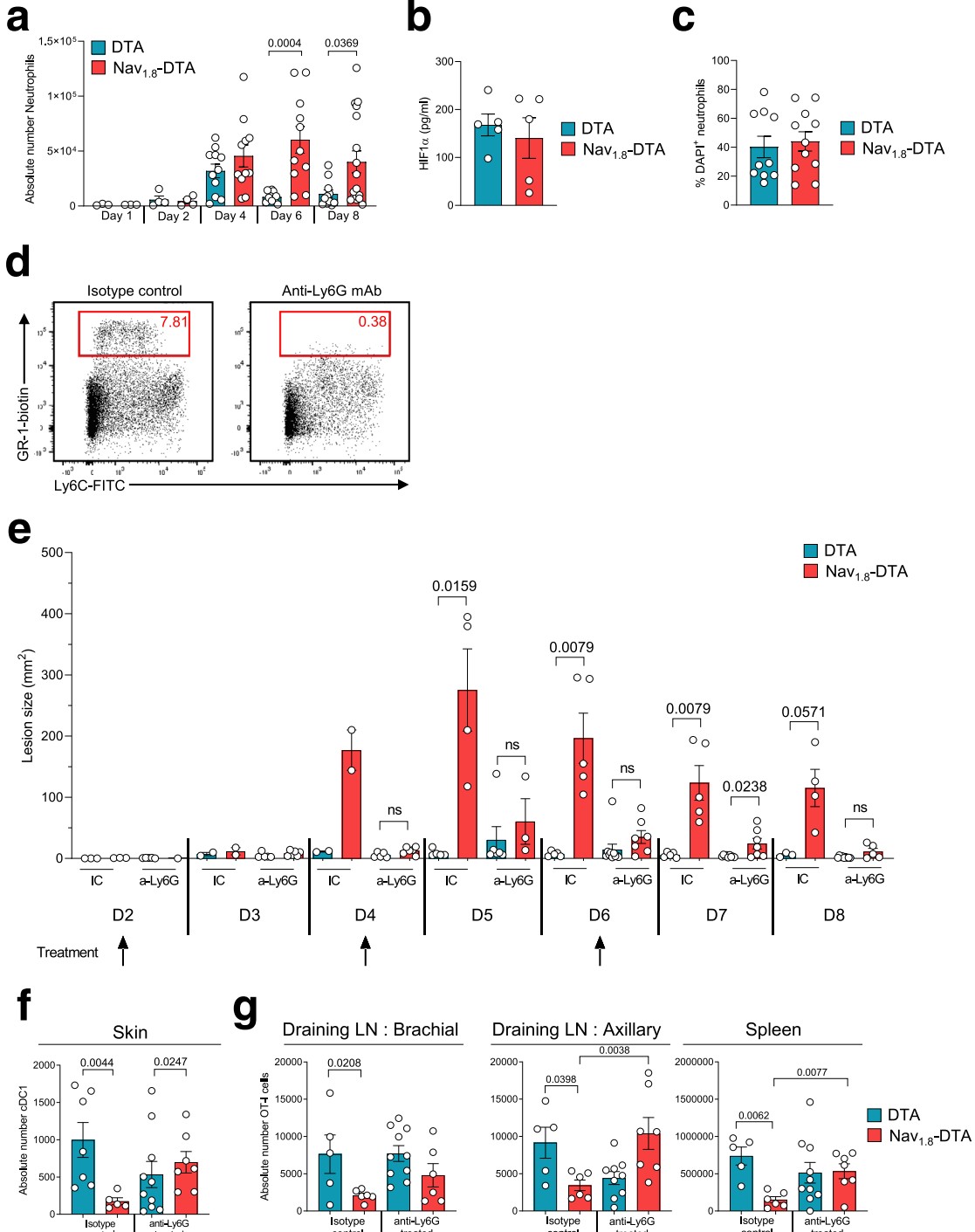

associated with higher levels of chemokines and inflammatory cytokines, including CXCL1, which is known to attract neutrophils to tissues[38]. Consistent with these data, neutrophil numbers in the skin remained higher in nociceptor-deficient mice than in their control DTA littermates, suggesting that this persistent neutrophil influx plays an important role in the observed extensive skin damage. This possibility was confirmed by our data showing that neutrophil depletion abolished the difference in the size of skin lesions between nociceptor-deficient and -sufficient mice.

Neutrophils are principally involved in the inflammatory response and pathogen clearance by phagocytosis and the degranulation of reactive oxygen species. Neutrophils have also recently been reported to help shape the adaptive immune response in the context of *Mycobacterium bovis* BCG inoculation, *Staphylococcus aureus* and influenza virus infection[30,31]. In the context of cutaneous HSV-1 infection, a previous study showed that neutrophils were dispensable for T cell priming[39]. Our data are consistent with this previous report, as we did not observe any difference in the magnitude of the CD8 T cell response in the skin, the dLN and the spleen upon neutrophil depletion in control DTA mice. However, we found that neutrophil depletion restored a normal T cell response in the dLN and spleen of nociceptor-deficient mice. The excessive neutrophil influx in the absence of

**Fig. 6 Regulation of the neutrophil response by nociceptors is required to reduce the severity of HSV-1-induced skin lesions and to elicit a robust CD8 T cell response. a** Longitudinal follow-up of the absolute numbers of neutrophils in the skin of HSV-1-infected control DTA and $Nav_{1.8}$-DTA mice. $N = 3$–17 mice per group. The data are presented as mean ± SEM. **b** HIF-1α concentrations in skin homogenates from control DTA and $Nav_{1.8}$-DTA mice 6 days pi. $N = 5$ mice per group. The data are presented as mean ± SEM. $P$ values were obtained by using a Mann–Whitney test (two-tailed). **c** Percentage of DAPI$^+$ neutrophils in the skin of HSV-1-infected control DTA and $Nav_{1.8}$-DTA mice 6 days pi. $N = 10$–11 mice per group. The data are presented as mean ± SEM. $P$ values were obtained by using a Mann–Whitney test (two-tailed). **d, e, f, g** Control DTA (blue) and $Nav_{1.8}$-DTA (red)-recipient mice were injected with 5 × $10^4$ naive virus-specific T cells (CD45.1$^+$ OT-I T cells) 1 day before HSV-OVA-TK$^-$ infection by flank scarification. Neutrophils were depleted using anti-Ly6G monoclonal antibody treatment. **d** Representative dot plots showing CD45$^+$ cells in the spleen of control (isotype control antibody-treated mice, left) and anti-Ly6G antibody-treated mice (right). Ly6C$^+$GR1$^{high}$ cells (gated area) were identified as neutrophils. **e** Size of skin lesions in infected control DTA and $Nav_{1.8}$-DTA mice treated with isotype control (IC) or anti-Ly6G antibodies. Arrows represent the time points at which anti-Ly6G or IC antibodies were injected ($N = 2$–10 mice per group). Data are presented as mean ± SEM. $P$ values were obtained by using a Mann–Whitney test (two-tailed). **f** Skin DCs were identified by flow cytometry 8 days pi (see the gating strategy presented in Fig. 3b). The absolute numbers of cDC1 in the skin are shown. Data are presented as mean ± SEM. $P$ values were obtained by using a Kruskal–Wallis test followed by Dunn's multiple comparison test. $N = 5$–10 mice per group. **g** Absolute numbers of OT-I cells in brachial LN, axillary LN and spleen of control DTA and $Nav_{1.8}$-DTA 8 days pi are shown. Data are presented as mean ± SEM. $P$ values were obtained by using a one-way ANOVA followed by Sidak's multiple comparison test. $N = 5$–10 mice per group.

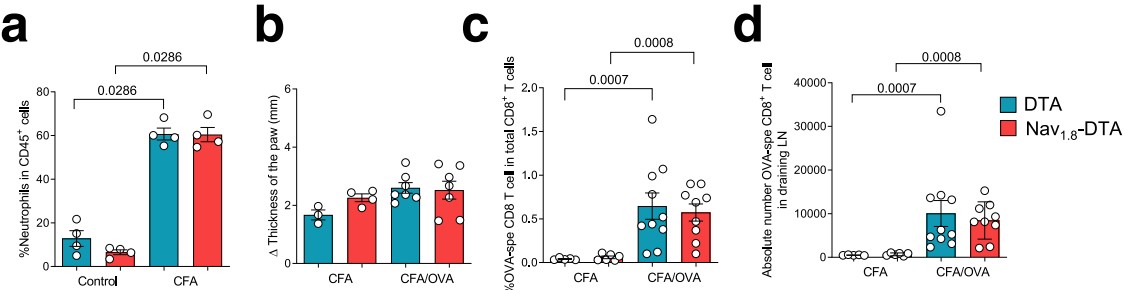

**Fig. 7 T cell and neutrophil responses are unaffected in $Nav_{1.8}$-DTA mice in a model of cutaneous vaccination. a** Percentages of neutrophils among CD45$^+$ cells within the skin foot pad 5 days after the injection of an emulsion of 50% PBS and 50% complete Freund adjuvant (CFA). $N = 4$ per group. **b–d** Injection of OVA (100 μg) in CFA into the paw of the mice. **b** Increase (Δ) in paw thickness relative to baseline 5 days after CFA or CFA + OVA injection. $N = 3$–7 mice per group. **c, d** Percentage of OVA-specific CD8$^+$ T cells among total CD8$^+$ T cells (**c**), $N = 5$–10 mice per group; and absolute numbers of OVA-specific CD8$^+$ T cells (**d**) in the skin-draining lymph node 7 days after the injection of CFA +/− OVA, $N = 5$–10 mice per group. **a–d** The results shown are from two independent experiments. The data shown are the mean ± SEM. $P$ values were obtained by using a Mann–Whitney test (two-tailed).

nociceptors is thus responsible for the defective primary T cell response. Importantly, this defect may have major consequences in terms of the efficacy of the antiviral response during the second phase of infection, when the virus replicates in DRG neurons and re-infects the skin. However, we were unable to investigate this issue because, in these sensory neuron-deficient mice, we had to use a TK-deficient virus that does not replicate in neurons and is therefore unable to generate a secondary skin infection.

Previous studies have suggested that neutrophils may inhibit T cell responses in viral infections by inhibiting T cell proliferation and inducing T cell apoptosis[31]. For example, it has recently been shown in human hepatitis B virus (HBV) infection, that the recruitment of neutrophils to the liver limits the immune response by inhibiting bystander and HBV-specific T cells in an arginase-dependent manner[32]. Other studies have also suggested that neutrophils can both negatively and positively affect antigen presentation under several conditions[33,34,40,41]. During the primary phase of replication in HSV-1 infection, the virus remains largely localised within the tissue and antigen transport by DC migrating from the skin to the LN is required to prime T cells[24,42]. In the dLN, virus-specific CD8 T cells do not interact directly with these migratory DCs, but their activation requires antigen cross-presentation by LN-resident cDC1[25]. We observed that the lower levels of virus-specific T cell expansion in the dLN of nociceptor-deficient mice were correlated with the presence of fewer cDC1 and LC in the skin of infected $Nav_{1.8}$-DTA mice. We also found that neutrophil depletion restored both the cDC1 and CD8 T cell responses. These data suggest that early events in the skin following HSV-1 infection are regulated by nociceptive

sensory neurons, affecting the quality and efficacy of T cell priming by antigen-presenting cells in the dLN. Consistently, purified DC from the draining LN of infected $Nav_{1.8}$-DTA mice were unable to prime naive T cells efficiently ex vivo. However, these cells were able to induce efficient T cell proliferation when incubated with the cognate peptide, suggesting that viral antigen uptake/processing by these DC, and/or DC migration to LN, is affected in this model. Along these lines, it has been shown that the catalytic activity of the neutrophil myeloperoxidase can inhibit adaptive immunity by suppressing DC activation, antigen presentation and migration to LNs[43].

Surprisingly, a defective CD8 T cell response was observed in the dLN and the spleen, but not in the skin of nociceptor-deficient mice. However, previous studies have shown that local inflammation in the skin is, by itself, sufficient to increase the recruitment of effector CD8 T cell populations[44]. It is, thus, possible that, despite the weaker systemic T cell response in the absence of nociceptors, the stronger cutaneous inflammation observed in these mice increases the level of T cell recruitment locally, compensating for the systemic defect. However, as the CD8$^+$ T cell counts in skin lesions do not appear to be affected by the lack of nociceptive neurons or neutrophils, the oversized lesions in $Nav_{1.8}$-DTA mice are probably induced by T cell-independent mechanisms.

Our findings are also consistent with recent studies describing a role for nociceptors in regulating neutrophil recruitment and activation in other infectious contexts[8,37]. In models of bacterial infection or arthritis, nociceptors have been shown to decrease the recruitment of neutrophils to sites of infection or to inflamed

joints[9,37,45,46]. By contrast, in other pathological conditions, nociceptors have been found to display pro-inflammatory properties, inducing the recruitment of neutrophils and other myeloid cells in the skin[10]. Another recent study reported that the optogenetic activation of cutaneous TRPV1[+] nerves alone is sufficient to trigger local inflammation associated with a cutaneous neutrophilic infiltrate[47]. Thus, in different contexts, nociceptive neurons can have very different effects on the neutrophil response and inflammatory processes. Our data are consistent with this notion, as the role of $Nav_{1.8}^{+}$ nociceptors in neutrophil and T cell responses observed in the model of cutaneous HSV-1 infection was not observed in a model of skin vaccination with CFA as an adjuvant. The role of these neurons thus depends strongly on the inflammatory conditions. In the CFA model, pain signals are perceived by the CNS in control mice but not in $Nav_{1.8}$-DTA mice[16]. However, no modulation of the neutrophil and T cell responses is observed. This strongly suggests that nociception per se (perceived at the central level) is not sufficient to induce the neuroimmune regulation observed during HSV-1 infection. These data, therefore, favour a model in which the effect of the sensory neurons is local in this infectious model.

The precise molecular mechanisms underlying the regulatory roles of neurons in the skin remain poorly characterised. The expression of the gene encoding $Nav_{1.8}$ (Scn10a) is restricted to peptidergic and nonpeptidergic neurons with a small diameter. This gene is not expressed in the large neurons known to express neurofilament heavy chain (NF200), a marker of A-fibre-associated sensory neurons[48]. Therefore, as previously described[16], in $Nav_{1.8}$-DTA mice, the majority (>85%) of peripherin-positive neurons are killed. These neurons include almost all isolectin B4 (IB4)-positive neurons and ~88% of calcitonin gene-related peptide (CGRP)-positive neurons. By contrast, the number of NF200[+] cells is only slightly reduced (~13%). Many neuropeptides and mediators, such as CGRP, substance P, pituitary adenylyl cyclase-activating polypeptide (PaCaP), prostaglandins and nitric oxide (NO) can be secreted by these nociceptive sensory neurons following their activation and have been implicated in neurogenic inflammation[9]. Further investigations are thus required to determine the precise role and cellular targets of each neuropeptide and mediator released in the skin upon sensory neuron activation in various inflammatory contexts, including HSV-1 infection. Studies in other genetic models in which subsets of nociceptive fibres can be depleted will be interesting. Nevertheless, this study reveals that sensory neurons involved in pain sensitivity play an important role in regulating the immune response to HSV-1 infection. This regulation is required to promote tissue repair after viral clearance. We also found that nociceptive sensory neurons downregulated the early innate inflammatory cytokine and chemokine responses, the activation of monocytes and the influx of neutrophils into the skin. Moreover, this regulation had a major impact further downstream, on the T cell adaptive immune response to HSV-1, paving the way for new avenues of research into treatments for cutaneous infections.

## Methods
**Mice**. $Nav_{1.8}$-Cre mice were kindly provided by Dr. Aziz Moqrich, with the permission of Prof. John Wood from UCL. Heterozygous $Nav_{1.8}$-Cre mice were crossed with homozygous DTA mice (B6.129P2-Gt(ROSA)26Sortm1(DTA)Lky/J The Jackson Laboratory JAX: 009669) to generate a 1:1 ratio of $Nav_{1.8}$-Cre-DTA and DTA littermates, as described[16]. OT-I mice (ovalbumin-specific TCR-transgenic mice)[26] were provided by Dr. Bernard Malissen. All the mice used have a C57BL/6 background and were bred and maintained under specific pathogen-free conditions at the Centre d'Immunophenomique (Ciphe) de Marseille and the Centre d'Immunologie de Marseille-Luminy (CIML). $Nav_{1.8}$-CrexRosa26-TdTomato ($Nav_{1.8}$-Cre-TdTomato) mice were obtained by crossing $Nav_{1.8}$-Cre mice with Rosa26-TdTomato mice (B6.Cg-Gt(ROSA)26Sortm14(CAG-tdTomato)Hze The Jackson Laboratory JAX: 007914). Mice were housed under a standard 12-h:12-h light–dark cycle with ad libitum access to food and water, 22 °C +/− 1 °C, 45–60% humidity. Age-matched (6–12-weeks old) and sex-matched (all the mice used were

female) littermate mice were used as controls. Experimental and control animals were housed and bred together. All experiments were conducted in accordance with institutional committees and French and European guidelines for animal care. Permission was granted to perform animal experiments by the institutional committee (Comité d'éthique en experimentation animale n°014) (project number: E13-055-10).

**Viruses and viral infection**. The thymidine kinase-deficient HSV-1-KOS.Cre transduced with ovalbumin (HSV-OVA-TK[−]), kindly provided by Dr. Francis Carbone, was grown and titrated on Vero cells (CSL) in minimal essential medium containing 10% FCS, 50 µM 2-mercaptoethanol, 2 mM L-glutamine, 100 U/ml penicillin and 100 µg/ml streptomycin (complete medium). The parental HSV-OVA has an expression cassette containing eGFP that includes a polytope with the sequence SIINFEKL-KA-TSYKFESV-KA-SSIEFARL at its carboxy terminus, under the control of the CMV-IE promoter. This cassette is inserted into the intergenic space between UL3 and UL4[49]. The resulting virus has a similar growth rate in vitro to the parental strain, replicates in skin and ganglia and the number of viral genomes detected during latency is similar to that for the wild type. The HSV-OVA-TK[−] virus was generated by co-transfecting cells with DNA isolated from the HSV-OVA virus and ScaI-cleaved pCP6277. The pCP6277 plasmid contains the SV40 terminator sequence inserted in the antisense orientation into the SstI site of pTK1[50]. TK viruses were selected on the basis of their resistance to acycloguanosine, as previously described[51], and disruption of the TK gene was confirmed by PCR. The HSV-OVA-TK[−] virus was plaque-purified twice before the production of a viral stock for inoculation, by combining the cell-associated and supernatant viruses collected from Vero cells infected at a low multiplicity of infection 3 days earlier. Mice were inoculated with HSV-OVA-TK[−] by flank scarification, as previously described[14,15]. Female mice, 6–12 weeks old, were anaesthetised by the i.p. injection (10 µl/g of body weight) of ketamine (2%)/Rompun (5%) solution in saline. The left flank of each mouse was clipped and depilated with Veet hair remover cream. A small area of skin, near the top of the spleen, was abraded with a MultiPro power tool (Dremel, Racine, WI) with a grindstone attachment (3.2 mm), held on the skin for 20 s to create a 2–4-mm$^2$ area of abraded skin. We then applied a 10-µl volume of viral suspension, containing 10$^6$ PFU, to the abraded skin and rubbed it in with a cotton-tipped applicator soaked in HBSS. A 1 × 2-cm piece of OpSite Flexigrid (Smith & Nephew, Hull, UK) was placed over the inoculation site to contain the virus during the initial infection. The flank of the mouse was wrapped in Micropore tape and then Transpore tape (3 M Health Care, St. Paul, MN) to prevent removal of the OpSite Flexigrid and disruption of the viral infection. The tape and Flexigrid were removed 48 h after infection.

In some experiments, mice received intravenous injections of 50,000 OT-I T cells and were infected, 1 day later, with 1 × 10$^6$ plaque-forming units of HSV-OVA-TK-, to study the virus-specific T cell response.

**Skin lesion analysis**. The surface of skin lesions surrounding the scarification site was measured from photographs using ImageJ 1.53c software. All the analyses of the lesion size were done by the same investigator that was blinded to the genotype of the mice during the analysis.

**Preparation of OT-I T cells**. T cells were isolated and purified from LN and spleens of transgenic OT-I mice by negative selection with the Pan T Cell Isolation Kit II from Miltenyi Biotec, according to the manufacturer's instructions. We depleted the cells other than transgenic CD8 T cells, by incubating them with biotin-conjugated antibodies against CD11b, CD11c, CD19, CD45R (B220), CD49b (DX5), CD105, MHC-class II and Ter-119. The antibody-labelled cells were removed by coupling to magnetic beads and passage through LS columns. Cell preparations were routinely 85–95% pure, as shown by flow cytometry.

**Measurement of viral titres**. Mice were infected with 1 × 10$^6$ plaque-forming units of HSV-OVA-TK[−] (strain KOS) by flank scarification. The primary inoculation site, defined as a 1.5 × 2-cm full-thickness piece of skin encompassing the scarified area, was excised and homogenised. No secondary lesions were observed as the thymidine kinase-deficient HSV strain was used. The titre of infectious virus in the tissue was measured in a standard assay determining the number of plaque-forming units on confluent Vero cell monolayers (CSL), as previously described[14].

**Immunisation with OVA and CFA**. Mice received 20 µl of OVA (100 µg) in an emulsion of 50% PBS and 50% complete Freund adjuvant (CFA) by intraplantar injection in the hind paw using a 25-µl syringe (Hamilton Co). The skin and popliteal draining LN were harvested for analysis. The OVA-specific CD8 T cell response was assessed with an H2-K$^{(b)}$-SIINFEKL-PE (OVA257–264 peptide) pentamer.

**Cytokine levels in tissues**. Skin samples (1.5 × 2-cm full-thickness pieces) were placed in lysing matrix tubes (MPBio) with 1 ml of lysis buffer (37% HCl, 2 mM EDTA) and mechanically dissociated with a three-cycle programme of FastPrep-24 5 G (MPBio). After centrifugation at 16,000 × g for 30 min, the supernatants were passed through a filter with 100-µm pores (Startedt) and stored at −80 °C until use.

Skin cytokine levels were determined with mouse IL-1β, IL-6, TNF-α, CCL2, CCL3, GM-CSF and CXCL1 cytometric bead array (CBA) flex set kits (BD Biosciences, Belgium), in accordance with the manufacturer's instructions. Skin IL-17A and IL-23 levels were determined with mouse IL-17A and mouse IL-23 cytometric bead array (CBA) flex set kits (BD Biosciences, Belgium). For these cytokines, the concentration was below the detection limit of the test (below 10 pg/ml) for all the samples. Briefly, capture beads were mixed and added to the test samples and serially diluted cytokine standards, which were then incubated for 1 h at room temperature. PE-labelled detection antibody was added to the samples, which were incubated for 1 h at room temperature and washed with the wash buffer. Samples were analysed on a BD LSR II flow cytometer with FCAP Array TM Software V3 (BD Bioscience) to determine the cytokine concentrations of the experimental samples. Mouse IFN-β was determined by ELISA (Ozyme, LEGEND MAX™ Mouse IFN-β ELISA Kit with pre-coated plates, Biolegend) according to the manufacturer's specifications. Mouse HIF-1α concentration was determined by ELISA (R&D Systems, Human/Mouse Total HIF-1 alpha DuoSet IC ELISA) according to the kit manufacturer's specifications.

**Immunofluorescence staining of the skin**. Mice were perfused with 10 ml PBS, and their ears were collected. The dorsal and ventral layers of the ear were separated. Skin samples were fixed by incubation for 1 h with 4% PFA in PBS, washed, permeabilised and saturated with a solution of 3% BSA, 0.2% Triton X-100 and 10% donkey serum in PBS. For staining of the epidermis, we separated the epidermis and dermis by incubating samples in 0.2 mg/ml dispase (GIBCO) for 2 h at 37 °C. The samples were then permeabilised and saturated by incubation in 3% BSA, 10% donkey and goat serum and 0.1% Triton X-100 in PBS for 1 h. The skin was stained by overnight incubation at 4 °C with the following primary antibodies: goat anti-mouse CD45 (purified, AF114-SP, R&D systems, 1/300), rat anti-mouse langerin (purified, eBioRMUL.2, 15237307, Fisher Scientific, 1/300), rabbit anti-mouse PGP9.5 (purified, RB-9202-P1, Fisher Scientific, 1/300) and hamster anti-mouse CD3 (APC, 145-2C11, 1000322, Biolegend, 1/300). The skin was washed several times and then incubated with the following secondary antibodies for 45 min at room temperature: goat anti-rat-488 (AlexaFluor 488, 112-545-003, Jackson Immunoresearch, 1/500), donkey anti-rabbit-555 (AlexaFluor 555, A31572, Invitrogen, 1/500) and donkey anti-goat-647 (AlexaFluor 647, 705-605-147, Jackson Immunoresearch, 1/500). The skin samples were washed again and mounted in a mounting medium on coverslips. Images were acquired with a Zeiss LSM780 confocal microscope and analysed with ZEN 2.3 software.

**Tissue dissociation**. Skin samples ($12 \times 12$-mm punch biopsy full-thickness pieces) were cut into small fragments and incubated in a collagenase/dispase/DNase digestion solution (0.2 mg/ml collagenase type IV (Sigma), 0.2 mg/mL dispase (GIBCO) and 1 mg/mL DNase (Roche)) in RPMI 1640 complete medium for 1 h at 37 °C. Then tissues were dissociated with 5-ml syringes fitted with 18 G needles and the resulting suspension was filtered on a cell strainer with 100-μm pores (BD Biosciences). The cells were washed with FACS buffer (5 mM EDTA in PBS 1X) to obtain a homogeneous cell suspension. Lymph nodes and spleen were crushed in FACS buffer on a cell strainer with 70-μm pores (BD Biosciences) and red blood cells from the spleen were lysed in RBC lysis buffer (Invitrogen). The suspensions were washed in FACS buffer, before filtering and staining for FACS analysis. For DRG neuron isolation, following intracardiac perfusion with 5 ml of 10× HBSS (magnesium- and calcium-free), 5 mM HEPES, 12.5 mM D-glucose and 1% penicillin/streptomycin, DRGs were carefully extracted and digested twice with 0.2 mg/ml collagenase type II (GIBCO) and 0.5 mg/ml dispase (GIBCO) for 30 min at 37 °C. The DRGs were washed several times with neurobasal complete medium (2% B-27, 20 mM L-glutamine and 10% penicillin/streptomycin) and then mechanically dissociated with three needles of decreasing diameter (18 G, 22 G and 26 G). The DRGs were then filtered on a cell strainer (70-μm pores, Miltenyi Biotec) and subjected to density gradient centrifugation through Percoll (12.8 and 28% Percoll in Leibovitz-15 complete medium supplemented with 5% FCS and 1% penicillin/streptomycin) to eliminate cell debris. The DRGs were washed several times with a neurobasal medium before staining.

**Flow cytometry**. Cells were incubated with Fc blocking antibody (2.4G2 from BD, 553141, 1/200) and with a fixable blue dead-cell staining kit (Invitrogen). Surface molecules were stained by incubation for 1 h at 4 °C with antibodies against CD11b (BV510, M1/70, 562950, 1/800), Ly6C (BV421, AL-21, 562727, 1/300), CD4 (FITC, H129.19, 553651, 1/100), CD11c (BUV395, N418, 744180, 1/200), CD24 (BUV737, M1/69, 612832, 1/1000), TCRβ (BV711, H57–597, 563135, 1/100), CD8 (APC, 53–6.7, 561093, 1/300 or APC-Cy7, 53-6.7, 560182, 1/100), CD19 (PE-CF594, 1D3, 562291, 1/200), NK1.1 (PE-CF594, PK136, 562864, 1/100), Ly6G (PE-CF594, 1A8, 562700, 1/300), Siglec F (PE, E50-2440, 562068, 1/500), Va2 (FITC, B20.1, 553288, 1/700) and γδ TCR (PE-Cy5, GL3, 15-5711-82, 1/500) all from BD. CD45 (BV785, 30-F11, 103149, 1/600), CD64 (BV711, X54-5/7.1, 139311, 1/300), c-kit (BV605, ACK2, 135122, 1/300), XCR1 (biotin, ZET, 148212, 1/200), GR-1 (Biotin, RB6-8C5, 108404, 1/500), Ly6G (APC-Cy7, 1A8, 127624, 1/500), MHC-II (AF700, M5/114.15.2, 107622, 1/600) and CD206 (APC, CO68C2, 141708, 1/400) all from biolegend. CCR2 (AF647, 475301, FAB5538R, 1/25) from R&D and CD3 (B610/20145-2C11, 1/300) from eBiosciences. For intracellular staining, cells were stained with antibodies against TNFα (PE, MP6-XT22, BD, 561063, 1/300), IL-6 (PE, MP5-20F3, BD, 554401, 1/100) and IL-1β (PE, NJTEN3, 4330860, BD, 1/300). Stained samples were analysed in a BD LSR Fortessa X20 flow cytometer (BD Biosciences).

**tSNE analyses**. tSNE analyses were performed with FlowJoTM version 10 software (FlowJo LLC), with 210,000 CD45+ cells from each mice genotype (control DTA and Nav1.8-DTA). In each group, the same number of CD45+ cells was used, to ensure an equivalent contribution of each mouse to the analysis. This equal cell number of cells was selected with the DownSample plugin. The CD45+ cells from the various mice were then merged with the concatenate tool and barcoded to track them and to distinguish the mouse strain. Finally, tSNE analyses were performed with the CD11c, MHC-II, CD206, CD11b, Ly6G, CD24, c-kit, CD4 and γδ TCR markers. The colours in the expression level heatmaps represent the median intensity values for a given marker. A four-colour scale was used, with blue, green–yellow and red indicating low, intermediate and high expression levels, respectively.

**T cell proliferation assay**. Control DTA and Nav1.8-DTA mice were infected with HSV-1-OVA-TK− by flank scarification. After four days, the brachial and axillary draining LN were collected and CD11c+ cells were sorted with the CD11c Microbeads UltraPure mouse kit from Miltenyi Biotec, according to the manufacturer's instructions. CD11c+ cells were incubated in a 96-well plate for 1 h at 37 °C with or without the OVA peptide (SIINFEKL). T cells were isolated and purified from the LN and spleens of transgenic OT-I mice by negative selection with the CD8α+ cell Isolation Kit from Miltenyi Biotec, according to the manufacturer's instructions, and stained with the Cell Trace Violet Cell Proliferation Kit, according to the manufacturer's instructions. CD11c+ cells and T cells were co-cultured (ratio of 1:10) for three days. Cells were stained with a fixable blue dead-cell staining kit (Invitrogen) and an anti-mouse CD8 antibody (PE-Cy5, 53-6.7, 55304 BD). T cell proliferation was assessed by flow cytometry.

**Neutrophil depletion**. DTA and Nav1.8-DTA mice received intraperitoneal injections of rat anti-mouse Ly6G monoclonal antibody, clone 1A8 (200 μg, BP0075-1, BioXCell), to deplete neutrophils. Injections began on day 2 post HSV-OVA-TK− infection and were performed every 2 days to ensure sustained neutrophil depletion. Control mice received an equivalent dose of the rat IgG2a isotype control (clone 2A3, BE0089, BioXCell). On day 8 post HSV-OVA-TK− infection, mice were killed and neutrophil numbers were analysed in the spleen, axillary LN, brachial LN and skin by flow cytometry.

**Statistical analysis**. All the mice used were female, and were assigned to experimental groups according to age. Statistical analysis was performed with Graphpad Prism 8 Software. Normality was tested with the Shapiro–Wilk test. Unpaired two-tailed Student's $t$ tests were used for comparisons if the data followed a Gaussian distribution with equal variances. Mann–Whitney $U$ tests were performed if this was not the case. One-way ANOVA was used for multigroup comparisons. Differences were considered significant for $P$ values < 0.05.

**Reporting summary**. Further information on research design is available in the Nature Research Reporting Summary linked to this article.

## Data availability

The experiment data that support the findings of this study are available from the corresponding author upon reasonable request. All data supporting the findings of this study are found within the paper and its Supplementary Information. Source data are provided with this paper.

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

## Acknowledgements

We thank Justine Galluso for mouse breeding and genotyping. We thank Mélanie Gabriac for the preliminary work performed during the initial phase of this project. We thank the Centre d'Immunologie de Marseille-Luminy (CIML) mouse house and core cytometry facilities. This project received funding from the European Research Council (ERC) under the European Union's Horizon 2020 research and innovation programme, under grant agreement No. 648768, from the *Agence Nationale de la Recherche* (ANR) (No. ANR-14-CE14-00 09-01), and the *Fondation pour la Recherche Médicale* (FRM). This work was also supported by institutional grants from INSERM, CNRS, Aix-Marseille University and Marseille-Immunopole to the CIML.

## Author contributions

J.F. and A.R. designed and performed experiments and analysed the data, L.Q., E.W., G.H., R.R., C.D. and G.D. helped in performing some experiments, S.U. conceived of, designed and directed the study, J.F. and S.U. wrote the paper and all authors reviewed and provided input on the paper. J.G. did the tSNE analysis. S.H. and B.M. provided important guidance for the characterisation of macrophage and DC subsets. C.M.J., L.K.M. and F.R.C. generated the HSV-OVA-TK⁻ virus and advised for the establishment of the infectious model. A.M. provided the Nav1.8-DTA mice as well as important guidance for the characterisation of sensory neurons. All authors contributed to the revision of the paper.

## Competing interests

The authors declare no competing interests.
