## [Peer Review File · Nature Communications]

Editorial Note: Parts of this peer review file have been redacted as indicated to remove third-party material where no permission to publish could be obtained.

Reviewers' comments:

Reviewer #1 (Infection immunity)(Remarks to the Author):

In their manuscript entitled 'Nociceptive sensory neurons promote the CD8+ T-cell response to HSV-1 infection', Filtjens et al. report that nociceptive neurons control antiviral CD8+ T cell priming and expansion by limiting neutrophil accumulation and subsequent cDC1 function. The study utilizes an elegant combination of a mouse model, NAV1,8-DTA, constitutively lacking nociceptive neurons and a modified HSV-1 which only replicates in keratinocytes, allowing the authors to exclusively study early events related to the primary phase of viral replication. The authors show that neutrophil depletion affects the numbers of antigen-specific T cells and cDC1 although the mechanism is not further elucidated and other immune cell subsets are likely to play a role in this phenomenon. Although several previous studies suggest a role of nociceptors in regulating early innate immunity in skin and lung, and, more specifically, the regulation of neutrophils, the reported observations are intriguing. Several aspects should, however, be clarified and further explored:

Major criticism:

The authors use NAV1,8-Cre mice crossed to iDTA (expression of the diphtheria toxin subunit A), inducing the constitutive absence of NAV1,8+ cells. Biological networks are very plastic and can compensate lack of specific celltypes. It is therefore not clear, whether the reported effects are secondary of such compensation-driven (mal-) adaptation, or whether they reveal a physiologic function of NAV1,8+ nociceptors. To support this latter notion, the conditional manipulation of nociceptor function would be needed. This can be achieved in the skin by photoactivation-approaches, or, sticking to the authors model, by crossing the NAV1,8-Cre mice to iDTR (expression of the diphtheria toxin receptor), which would allow to deplete the NAV1,8+ cells by DT treatment just before, or during infection.

The study does not account for the lack of early IL23/IL17 dependent protective immunity triggered by nociceptors in the skin in response to infection, but also tissue stress alone. It is conceivable, that the scarification and local infection-induced skin damage induce bacterial infection in mice that lack the nociceptor-dependent regulation of IL23/IL17 dependent protective immunity. Is the increased neutrophil recruitment and persistence hence a consequence thereof? Importantly, neutrophil recruitment is observed even during PBS scarification of NAV1,8-DTA mice (Fig S1), suggesting that tissue damage or local bacterial infection are sufficient for the phenotype in these mice. These questions should be experimentally addressed and discussed.

A weak point of this study is that cDC1 are studied on d8 pi, when T cell priming has already taken place. To support the idea that altered numbers and/or function of cDC1 affects the antiviral T cell response in these mice it would be critical to study these cells early during infection (d1-3), which is when priming occurs in the LNs. More functional characterization would be required (maturation, per-cell T-cell stimulatory capacity) to even correlate a phenotype of cDC1 in these mice with T cell priming.

Specific Comments:

The study of HSV-1 in a setting where sensory neurons are absent is highly challenging and some care needs to be taken to specifically study the influence of nociception rather than the presence of the neurons themselves, which serve as replication spot for the virus. Employing other models used to

study anti-viral skin immunity (e.g. sensitization with vaccinia virus) would further strengthen the conclusions of this paper.

The kinetics of the accumulation of various cell types and the purportedly resulting tissue damage should be studied further. A significant difference in neutrophil numbers between NAV1,8-DTA mice and littermates can only be shown on day 8 p.i. whereas striking differences in lesion sizes and severity appear on day 3 p.i.. Both absolute and relative numbers of other cell populations reported to be involved in this phenomenon are only shown on day 8 p.i. when the macroscopic lesions are almost entirely resolved. The authors at times (e.g. abstract) imply a direct connection between the inability of the NAV1,8 to downregulate neutrophil recruitment and the DC response which should not be stated as such, given that these events do not appear to take place in this order. Rather, as is stated in the discussion, nociceptive neurons might affect early inflammatory stimuli, including neutrophil function or phenotype, which affect DC function and subsequent T cell expansion.

The severity of the lesions and their response to neutrophil depletion is not entirely consistent with the authors' hypothesis (see Figure 4B). Whereas the appearance and the peak of visible lesions in NAV1,8-DTA mice is delayed by 1-2 days upon neutrophil depletion, it appears that the maximal size of the lesions is unaltered and their clearance is not strikingly enhanced. This comparison however, is never explicitly made in the text or figures. Instead, the authors highlight that neutrophil depletion ablates the differences between NAV1,8-DTA and DTA mice which is supported by the data but appears to be due to the DTA mice having vastly increase lesion sized following the depletion. This seems not a commonly observed effect in HSV-1 infections (see e.g. Wojtasiak et al, 2010) and should be discussed further.

Importantly, as reported previously, nociceptors seem to act very early during infection and/or tissue damage also in this model, suggesting potential alternative explanations for the increased lesion and neutrophil influx (lack of early IL23/IL17 immunity, local bacterial infection?). These possibilities should be explored (see above).

The authors fail to establish a convincing link between the severity of the lesions and the reported neutrophil-cDC1-CD8+ T cell axis. The numbers of CD8+ T cells in the skin lesions appears to be unaffected by either lack of nociceptive neurons or neutrophil depletion and it remains unclear how the more severe lesions or the lack of tissue repair in the NAV1,8-DTA mice is caused.

The authors show accumulations of other immune cell subsets which could conceivably affect the course of HSV-1 infection and T cell priming / expansion as well. Monocyte numbers are decreased in mice lacking nociceptive neurons (Fig. 3B) as are cDC2 although the data is contradictory (significant in Fig 3D, non-significant in 4C). This should at the very least be discussed and does weaken the causal link the authors report.

Functional assessment of the immune populations in skin and lymph nodes might strengthen the hypothesis of the authors and provide an explanation for the increase inflammation and tissue damage beyond the numerical enrichment of immune cells.

Minor points

Fig 2c,d,: skin lesions vary a lot, how are skin samples taken, processed and normalized for pfu, and for measurement of cytokines and chemokines? Information is lacking also in M&Ms

Fig 3a/b: a tSNE figure is not necessarily helpful in this context; a clear figure showing gating strategy in combination with the quantification in 3c would be more helpful.

Fig 4b: The more relevant comparison and visualization would be NAV1,8-DTA with or without

neutrophil depletion in order to make a point about the role of neutrophils in inducing tissue damage.

Fig4d: Viral titer in this setup should be shown to ensure that increased numbers of XCR1+ DC and OT-1 T cells are not due to the mice failing to control the infection due to their lack of neutrophils.

S1b: It is unclear what was chosen as a 'starting weight' since the DTA mice (presumably the 'control mice' which are mentioned as being used to determine 'starting weight') already start the experiment at a 0.9 relative weight.

S2d: Editing error in axis name

S1A "after scarification with PBS", not "after infection"

Fig S4 (and others) statistics are lacking

S4B: FACS plots cannot be from same staining, CD24 levels/MFI are very different in the different plots

Reviewer #2 (Neuro-immune crosstalk)(Remarks to the Author):

This is an interesting study, which contains information regarding the effect/role of parenchymal cells of the nervous system in controlling HSV-1 processes. The authors demonstrated that the CD8 T-cell response to HSV-1 is impaired in nociceptor-deficient mice, which displayed enhanced skin inflammation upon HSV-1 infection. Their interpretation of the results as that the nociceptor is potentially a possible molecule involved in the interaction of T cells, which was supported by the observations that the nociceptor-deficient mice had an impaired CD8 T-cell response to HSV-1, and that lack of nociceptive sensory fibres in Nav1.8-DTA mice did not affect the viral clearance.

The paper would be more convincing if the following points can be addressed more clearly:

1. The study used a virus with a mutation of the TK gene that expressed the SIINFEKL-peptide derived from ovalbumin (OVA) and assessed the viral responses relied on responses of transferred OT-I T cells. It is not clear whether the intensity of OT-I responses completely parallel with the real viral responses of HSV-1 infection and whether this augmented responses is antigen-specific, or it is restricted to viral responses only.
2. It remained to be tested whether blockade of nociceptive effect could modulate intensity of HSV response. Such attempts should further support the predicted hypothesis.
3. The authors analyzed the virus-specific T-cell response in the skin draining lymph nodes by monitoring the expansion of the transferred OT-I T cells, wondering whether how the responses in the skin reflects the immune responses taken place in the CNS.
4. Mechanistic studies showed that cytokine/chemokine levels were higher in infected Nav1.8-DTA mice. However, such changes may just reflected a stronger immune response. It is not examined whether producing cells also involved parenchymal cells in addition to infiltrating immune cells, and whether the increased cytokine production was caused by over-infiltration of PMN or reduced monocyte-infiltration, which may also cause greater immune responses, due to that an over infiltration of monocytes is also inhibitory.

Overall the paper is clearly presented. Demonstration of a correlation of nociceptive and viral intensity is interesting; however, the mechanisms leading to augmented immune responses could be more sophisticated.

Reviewer #3 (HSV, viral immunity)(Remarks to the Author):

Filtjens and colleagues used an established murine skin scarification protocol to investigate the contribution of nociceptive sensory neurons and neutrophils to mount robust CD8 T cell priming and robust adaptive immune response against herpes simplex virus type 1 (HSV). This study provides further support to the emerging concept that the nervous system has important functions in addition to the immune system to control inflammation during an infection. The authors show that there was a defective primary CD8 T cell response against HSV leading to an exacerbated disease pathology associated with extensive skin lesions and an enhanced inflammatory cytokine and chemokine production in mice ablated for Nav1.8+ sensory neurons. Further experiments indicate that the neurons were required to downregulate the number of neutrophils to the infected skin, to control the skin XCR1 dendritic cell response, and to limit the tissue damage. Overall, the authors designed, controlled and executed their experiments well. However, important information is missing in some instances, which prevents the manuscript from being comprehensible to potential readers from the diverse fields of virology, neurobiology and immunology.

Major Comments

1. Introduction, page 3 and 4: Could the authors please specify what kind of mediators are locally released by nociceptive sensory neurons upon activation that could modulate the activity of immune cells? Which mediators have proinflammatory and which ones anti-inflammatory activity?
2. Introduction page 4 bottom, results page 5, discussion page 12/13, and/or Exp Procedures page 17: The authors need to provide a better introduction and characterization of the genetic mouse model used to ablate the nociceptor neurons. Neurons other than nociceptors express Nav1.8 in the mouse. Up to 75% of the neurons in dorsal-root ganglia express Nav1.8 Cre (Shields et al. 2012, Pain). The authors need to provide evidence, which neurons were lacking upon ablation in the skin and in the dorsal root ganglia; for example, by histology studies to characterize the neurons and neuronal processes that remain in the skin and DRG after ablation. Which skin cells in addition to the nociceptor neurons expressed Nav1.8+? Which immune cells expressed Nav1.8+? To what extent were such Nav1.8+ expressing cells ablated in this model? What is the genetic background of the mice used in this study?
3. Results page 5, discussion page 12, Exp Procedures page 17: The authors need to provide a better description of the HSV-1 mutant that they have used in this study. Which HSV strain was used as parental, and what is the strain history of the HSV-TKminus mutant? Where in the genome or in which protein has the OVA SIINFEKL epitope been integrated? How was the inoculum prepared and purified? Cell extract, extracellular medium, gradient purification?
4. Results page 7: Could the authors provide a rationale for their selection of cytokines that they have measured? What are the sources and the functions of IL-1beta, IL-6, TNFalpha, IFN-beta, GM-CSF, CXCL1, CCL3, or CCL2 in this infection model?
6. Results pages 7ff, Figure 2 and Figure 3: There were similar numbers of neutrophils in the skin at 4 dpi in Nav1.8+ and Nav1.8-ablated mice, while there were more neutrophils in the Nav1.8-ablated mice at 8 dpi. The authors concluded that more neutrophils had been recruited to the skin in the Nav1.8 ablated mice. However, an alternative hypothesis is that more neutrophils had been unable to leave the skin. Several groups have reported the phenomenon of neutrophil reverse migration, and suggested that the expression of hypoxia inducible factor 1 alpha might contribute to an increased

retention of neutrophils, which would also delay the resolution of inflammation. The authors could investigate the level of HIF-1 α in their system to address whether it contributes to neutrophil behaviour and inflammation. Furthermore, the high levels of cytokine expression in the skin in the absence of Nav1.8 may contribute more than the increased neutrophil numbers to the delayed resolution of inflammation.

8. The authors show the effects that ablating Nav1.8+ neurons have on cytokine expression, neutrophil levels in the skin, and CD8 T cell response against HSV. It would be interesting to know how activating Nav1.8 with agonists or with optogenetic methods would influence the immune system in this system.

9. Pro-inflammatory cytokines, like TNF, can sensitize sodium channels including Nav1.8. The authors need to discuss whether the high level of TNF in the Nav1.8-ablated mice could be the consequence of this lack of Nav1.8 sensitization.

Minor Comments

1. Discussion, page 12: Last sentence is unclear.

5. Exp procedures page 17: How were the flank scarification and HSV infection performed? Amount of virus, time of inoculation, and duration of infection?

Point by point response to the reviewer's comments:

Reviewer #1 (Infection immunity)(Remarks to the Author):

In their manuscript entitled 'Nociceptive sensory neurons promote the CD8+ T-cell response to HSV-1 infection', Filtjens et al. report that nociceptive neurons control antiviral CD8+ T cell priming and expansion by limiting neutrophil accumulation and subsequent cDC1 function. The study utilizes an elegant combination of a mouse model, NAV1,8-DTA, constitutively lacking nociceptive neurons and a modified HSV-1 which only replicates in keratinocytes, allowing the authors to exclusively study early events related to the primary phase of viral replication. The authors show that neutrophil depletion affects the numbers of antigen-specific T cells and cDC1 although the mechanism is not further elucidated and other immune cell subsets are likely to play a role in this phenomenon. Although several previous studies suggest a role of nociceptors in regulating early innate immunity in skin and lung, and, more specifically, the regulation of neutrophils, the reported observations are intriguing.

- We appreciate the positive feedback from the reviewer.

Several aspects should, however, be clarified and further explored:

Major criticism:

1) The authors use NAV1,8-Cre mice crossed to iDTA (expression of the diphtheria toxin subunit A), inducing the constitutive absence of NAV1,8+ cells. Biological networks are very plastic and can compensate lack of specific celltypes. It is therefore not clear, whether the reported effects are secondary of such compensation-driven (mal-) adaptation, or whether they reveal a physiologic function of NAV1,8+ nociceptors. To support this latter notion, the

conditional manipulation of nociceptor function would be needed. This can be achieved in the skin by photoactivation-approaches, or, sticking to the authors model, by crossing the NAV1,8-Cre mice to iDTR (expression of the diphtheria toxin receptor), which would allow to deplete the NAV1,8+ cells by DT treatment just before, or during infection.

- The Nav1.8-DTA model has been previously used in many studies to demonstrate a role of nociceptive sensory neurons in various biological functions (Abrahamsen et al., 2008; Chiu et al., 2013; Riol-Blanco et al., 2014; Talbot et al., 2015). We therefore believe that this genetic model, which is commonly used, is widely accepted as a model to study the role of sensory neurons expressing Nav1.8. Importantly, in steady state conditions, we did not detect any difference in the skin structure or in immune cell compartments between Nav1.8-DTA and DTA control mice. Moreover, we analyzed the immune response of Nav1.8-DTA mice in another model of skin inflammation (New Fig. 4g-j). In this other model, we did not detect any difference in the neutrophils and CD8 T cell responses (for more details, please see our response to the point 5 of the reviewer#1, below). These data show that the regulation of the neutrophil and T cell responses by sensory neurons is context-dependent. They also demonstrate that the neutrophil response is not intrinsically affected in Nav1.8-DTA mice, highlighting the relevance of our genetic model to study nociceptive sensory neurons in the context of HSV-1 infection.

We decided not to use a model in which Nav1.8 neurons express the diphtheria toxin receptor (iDTR) because this model requires the injection of a toxin that will induce neuronal cell death in the tissue, potentially inducing inflammatory reactions that could interfere with the processes we are studying.

2) The study does not account for the lack of early IL23/IL17 dependent protective immunity triggered by nociceptors in the skin in response to infection, but also tissue stress alone. It is conceivable, that the scarification and local infection-induced skin damage induce bacterial infection in mice that lack the nociceptor-dependent regulation of IL23/IL17 dependent protective immunity. Is the increased neutrophil recruitment and persistence hence a consequence thereof?

- We are using a well-established model of HSV infection in which, after virus inoculation the scarification site covered during two days to limit inadvertent bacterial infection. We have never detected overt bacterial infection after HSV scarification, nor the Carbone Laboratory whom established this model, even in immunodeficient hosts including RAG-KO mice (Van Lint et al., 2004). Moreover, the Nav1.8-DTA model has been used to study the role of sensory neurons in a model of psoriasis in which bacterial co-infections were not described either (Riol-Blanco et al., 2014). In addition, another study investigated the role of sensory neurons in the context

of *Candida albicans* infection (epicutaneous infectious model in which stratum corneum is removed with grit sandpaper before the application of *C. albicans*) (Kashem SW, et al., 2015). In this study, bacterial co-infections were not described in mice surgically denervated on one lateral side of their dorsum despite a reduction of IL-23 and IL-17 expression. Therefore, we think that the phenotype we observe in our mice is not due to bacterial co-infection. Nonetheless, to address the reviewer's question more specifically, we measured the levels of IL-17 and IL-23 in the skin of Nav_{1.8}-DTA and control DTA mice after scarification (PBS) or after HSV-1 infection. IL-23 and IL-17 were not detected in skin lysates. These data are now described in the manuscript, lines 154-155.

3) Importantly, neutrophil recruitment is observed even during PBS scarification of NAV1,8-DTA mice (Fig S1), suggesting that tissue damage or local bacterial infection are sufficient for the phenotype in these mice. These questions should be experimentally addressed and discussed.

- As mentioned above, HSV infection was performed in sterile conditions, and the scarification site were covered for 2 days after infection. None of these HSV lesions showed any apparent signs of bacterial superinfection. The data initially presented in the manuscript showing the lesion size in the scarification/PBS condition were based on a quite limited number of mice. Due to the COVID-19 crisis, Jessica Filtjens (1st author of this study) could not come back to the lab to perform the revision of the manuscript. Therefore, Anais Roger (PhD student) put the infection protocol back in place to perform the additional experiment requested. This allowed us to increase the number of animals included in these analysis. All the results obtained by the two experimenters regarding the monitoring of lesion size were reanalysed blindly from the photographs. They have been compiled and are now presented in the new Figures 1B, 4C, 4D and Supplemental Figure 1Ds. These additional analysis did not revealed any significant difference in lesion size between the two genotypes in the scarification/PBS condition.

These data are consistent with the additional figure 1 (below), which shows a comparison of neutrophil frequencies in the skin of PBS-treated (scarification) and HSV-1-infected mice in control DTA mice (black bars) and NAV1.8-DTA mice (white bars) at day 1, 2, 4 and 8 post-inoculation. Scarification without infection increased neutrophil frequency in the skin at day 4 post-lesion but no significant difference was observed between the two genotypes at any time point. The higher neutrophil infiltration in the skin of Nav1.8-DTA mice observed on days 6 and 8 pi (see new Figure 2C and 4A) is thus HSV-1-dependent.

Additional Figure 1 (for reviewers only): Percentages of neutrophils in CD45⁺ skin cells from HSV-1-infected or PBS-treated control DTA (black bars) and Nav1.8-DTA (white bars) mice.

The lack of lesion size difference between Nav1.8-DTA and control mice in scarification/PBS condition is also consistent with the fact that there is no difference in term of cytokine/chemokine production between the two genotypes in the absence of HSV-1 infection (Fig. 1D).

4) A weak point of this study is that cDC1 are studied on d8 pi, when T cell priming has already taken place. To support the idea that altered numbers and/or function of cDC1 affects the antiviral T cell response in these mice it would be critical to study these cells early during infection (d1-3), which is when priming occurs in the LNs. More functional characterization would be required (maturation, per-cell T-cell stimulatory capacity) to even correlate a phenotype of cDC1 in these mice with T cell priming.

- We thank the reviewer for this suggestion. We now provide additional analysis on DC responses in the skin and skin draining LN on days 2, 4 and 6 pi (New Figure 3B, new Supplemental Figures 2c, 3 a-c, 4a-c, 5a-d). The numbers of DC subtypes (cDC1, cDC2, LC) were similar in the skin of uninfected Nav1.8-DTA and DTA mice (Figure S4A, B, C, left panels). In HSV-1-infected mice, we did not detect any difference in DC counts on days 2 and 4 pi (Figure S4A, B, right panels). By contrast, and consistently with our previous data on day 8 pi (initial Figure 3), we observed that the absolute numbers of cDC1 and LC were reduced in the skin of Nav1.8-DTA mice on day 6 pi (new Figure 3B), confirming that the DC response is affected in the absence of nociceptive neurons.

Upon HSV-1 infection, skin DC migrate from the skin to the draining lymph nodes were they are able to cross-present viral antigens to CD8 T cells (Allan et al., 2006; Bedoui et al., 2009;

Eidsmo et al., 2009; Hor et al., 2015). Previous studies have shown that DC migrating from the skin can transfer antigens to LN-resident DC, facilitating T-cell activation (Allan et al., 2006; Hor et al., 2015). We did not detect any difference in the frequency of DC subsets in the skin draining LN of infected Nav1.8-DTA and control DTA mice (Figure S5A-D). However, as suggested by the reviewer, the regulation of the DC response in the skin of nociceptor-deficient mice could affect the quality of T-cell priming in the dLN. We therefore analyzed whether purified DC isolated from the skin draining LN of infected DTA and Nav1.8-DTA mice on day 4 pi were similarly able to prime naïve OT-I T cells in vitro. T cell proliferation was analyzed by measuring the decrease in the mean fluorescence intensity (MFI) of fluorescently-labelled T cells after 4 of co-culture. When the OVA peptide was added exogenously in the cell cultures, OT-I T cells proliferated similarly when they were incubated with DC from Nav1.8-DTA and control DTA mice (Figure 3C). By contrast, in the absence of exogenous peptide, DC from Nav1.8-DTA infected mice were unable to induce OT-I T cell proliferation at the level observed with control DC (Figure 3D). These new data show that the DC are affected in their capacity to present viral antigens in the draining LN in mice lacking Nav1.8⁺ sensory neurons.

5) Specific Comments:

The study of HSV-1 in a setting where sensory neurons are absent is highly challenging and some care needs to be taken to specifically study the influence of nociception rather than the presence of the neurons themselves, which serve as replication spot for the virus. Employing other models used to study anti-viral skin immunity (e.g. sensitization with vaccinia virus) would further strengthen the conclusions of this paper.

- We used a TK-deficient HSV-1 strain which does not replicate in neurons (Bedoui et al., 2009; Tenser, 1991). In this experimental model, the lack of nociceptive fibers does not directly affect viral replication as shown in Figure 1C. This strategy is therefore relevant to dissect precisely the immunoregulatory role of nociceptive sensory during the early phase of cutaneous infection with HSV-1.

Whatever the inflammatory or infectious model used, it will be very difficult to distinguish a potential role of nociception from a local role of the neurons themselves. Indeed, nociceptor peripheral nerve terminals possess receptors and ion channels that detect molecular mediators, released during inflammation (Baral et al., 2019). These include including ATP, bradykinin, histamine, growth factors and cytokines which are produced upon infection/injury. Sensory neurons activation and pain perception by the central nervous system should therefore be observed in most inflammatory contexts.

We know from previous studies that the immunoregulatory effects of the nervous system on the inflammatory response are highly context-dependent. The pro- or anti-inflammatory nature of these effects depends on the pathological condition (Baral et al., 2019; Chiu et al., 2013; Kashem SW, Riedl MS, 2015; Riol-Blanco et al., 2014; Talbot et al., 2015). The aim of our study was to investigate specifically the role of nociceptive sensory neurons in the context of cutaneous infection with HSV-1. We chose to study this viral infection because HSV-1 infects both skin epithelial cells and the nervous system, and is therefore a model of choice for studying neuroimmune interactions. Furthermore, this viral infection is highly prevalent in the human population. Therefore, studying the neural regulation of the immune response in this model, is in itself, interesting and important.

However, to address the reviewer's question, we analyzed another model of skin inflammation. Neutrophil and anti-OVA CD8 T responses were analysed in a skin vaccination model. We immunized mice with ovalbumin in complete Freund's adjuvant (CFA), an adjuvant commonly used to mimic a microbial infection and to induce a strong immune response. We used this model because previous studies demonstrated that the hyperalgesia associated with CFA injections is lost in Nav1.8-DTA mice (Abrahamsen et al., 2008), suggesting that Nav1.8-expressing neurons are activated and could therefore play a role after CFA-induced inflammatory insult.

Neutrophil influx upon CFA injection was similar in the skin of Nav1.8-DTA and DTA mice (new Figure 4G). Moreover, the level of skin inflammation analysed by measuring changes in the thickness of the mouse's paw was similar for both genotypes (new Figure 4H). Finally, we did not detect any difference in the frequency or absolute number of OVA-specific CD8 T cells in the draining LN of Nav1.8-DTA and DTA mice upon immunisation (new Figure 4I, J).

These data are very important for 3 reasons:

- 1) They demonstrate that the regulation of the neutrophil and T cell responses by Nav1.8⁺ sensory neurons is context-dependent.
- 2) In the CFA model, we know that pain signals are perceived by the CNS in control mice but not in Nav1.8-DTA mice (Abrahamsen et al., 2008). However, the modulation of the neutrophil and T response was not observed. This strongly suggests that nociception by itself at the central level is not sufficient to induce the neuroimmune regulation observed upon HSV-1 infection. These data therefore favour a model in which the effect of the sensory neurons in this model is local.
- 3) In line with our answer to the question 1 of the reviewer, these data also demonstrate that the neutrophil response is not intrinsically affected in Nav1.8-DTA mice, highlighting the relevance of our genetic model to assess functional role on Nav1.8⁺ neurons.

These new data are now described in the revised manuscript (lines 330-338) and discussed (lines 427-435) in details.

The kinetics of the accumulation of various cell types and the purportedly resulting tissue damage should be studied further. A significant difference in neutrophil numbers between NAV1,8-DTA mice and littermates can only be shown on day 8 p.i. whereas striking differences in lesion sizes and severity appear on day 3 p.i.. Both absolute and relative numbers of other cell populations reported to be involved in this phenomenon are only shown on day 8 p.i. when the macroscopic lesions are almost entirely resolved.

- This point is well taken. New data on the kinetic of the response have been added in the revised manuscript (see new Fig. 4A and new Suppl. Fig. 6A and B). We also analyzed in more details the skin immune response on day 6 when the cytokine response is affected. These new data are presented in the new Fig. 2, 3 and 4 and the new Supplementary Fig. 2, 3, 4.

The authors at times (e.g. abstract) imply a direct connection between the inability of the NAV1,8 to downregulate neutrophil recruitment and the DC response which should not be stated as such, given that these events do not appear to take place in this order. Rather, as is stated in the discussion, nociceptive neurons might affect early inflammatory stimuli, including neutrophil function or phenotype, which affect DC function and subsequent T cell expansion.

- We now provide additional data showing that the neutrophil response and the DC response are also affected in the skin on day 6 pi (new Fig. 2, 3B). This is consistent with the increased levels of inflammatory cytokines and chemokines in Nav1.8-DTA mice at this time point (Fig. 1). These phenotypes are thus present before the defective CD8 T cell response, which is observed on day 8 in these mice (Fig.3F, G).

Neutrophils were shown to affect antigen presentation and DC migration to the LN (Appelberg, 2007; Odobasic et al., 2013; Soehnlein et al., 2009). We therefore investigated whether the larger number of neutrophils in the skin of nociceptor-deficient mice was responsible for the defective CD8 T-cell response by analyzing the DC and T cell responses to HSV-1 infection in the skin of Nav1.8-DTA and DTA mice treated with the anti-Ly6G antibody (Fig. 4E, F). Consistently with the decreased frequency of cDC1 in the skin of Nav1.8-DTA mice 6 days pi (new Figure 3B), we also observed significant fewer cDC1 in the skin of nociceptor-deficient mice compared to littermate controls upon IC antibody treatment 8 days pi (Fig. 4E). This defect was reversed by neutrophil depletion with the anti-Ly6G antibody, suggesting that the down-regulation of neutrophil recruitment by nociceptive sensory neurons could have an impact on the cDC1 response in the skin.

We also investigated whether the impairment of the CD8 T-cell response in mice lacking nociceptive sensory neurons was linked with the dysregulated neutrophil response in these mice. Consistent with our previous data (Figure 3), in mice treated with the isotype control (IC) antibody, the OT-I T cell population expanded less in the dLN and spleen of infected Nav1.8-DTA mice than in those of DTA controls (Figure 4F). By contrast, neutrophil depletion rescued this phenotype and restored normal T-cell responses in both the spleen and the draining LN of Nav1.8-DTA (Figure 4F).

Therefore, as stated by the reviewer “nociceptive neurons might affect early inflammatory stimuli, including neutrophil function or phenotype, which affect DC function and subsequent T cell expansion.” We modified the abstract and the text of the manuscript to make it clearer.

The severity of the lesions and their response to neutrophil depletion is not entirely consistent with the authors’ hypothesis (see Figure 4B). Whereas the appearance and the peak of visible lesions in NAV1,8-DTA mice is delayed by 1-2 days upon neutrophil depletion, it appears that the maximal size of the lesions is unaltered and their clearance is not strikingly enhanced. This comparison however, is never explicitly made in the text or figures. Instead, the authors highlight that neutrophil depletion ablates the differences between NAV1,8-DTA and DTA mice which is supported by the data but appears to be due to the DTA mice having vastly increased lesion size following the depletion. This seems not a commonly observed effect in HSV-1 infections (see e.g. Wojtasiak et al, 2010) and should be discussed further.

- We would like to apologize because there were errors in the original Figure 4B. Additional analysis have been done which have allowed us to consolidate these data which are shown in new Figure 4C (see also our answer to point 3 above). These data show that neutrophil depletion was sufficient to drastically reduce the lesion size in infected Nav1.8-DTA mice. They also show that neutrophil depletion did not alter lesion size in infected control (DTA) mice which is consistent with the previous studies mentioned by the reviewer.

Importantly, as reported previously, nociceptors seem to act very early during infection and/or tissue damage also in this model, suggesting potential alternative explanations for the increased lesion and neutrophil influx (lack of early IL23/IL17 immunity, local bacterial infection?). These possibilities should be explored (see above).

- As mentioned previously, the increase in inflammatory cytokine and chemokine production in the skin of Nav1.8-DTA mice is observed after HSV-1 infection but not after scarification/PBS only (Figure 1D). If the Nav1.8-DTA mice would present an increased susceptibility to bacterial infection, it should also be observed in the scarification/PBS condition, which is not the case.

Moreover, as mentioned above, we did not detect any activation of the IL-23/IL-17 axis in these mice (this point is now stated in the paper lines 153-154). However, we described an increase in other inflammatory cytokine and chemokine levels in the skin of infected Nav1.8-DTA mice (**Figure 1D**). We therefore performed additional experiment to dissect the cellular source of the inflammatory cytokines IL-6, TNF- α and IL-1 β . We observed that the frequency of monocytes producing TNF- α and IL-1 β was increased at day 6 post-infection (new Figure 2). These data favor a model in which after the peak of viral replication, nociceptive sensory neurons down regulate monocyte activation and their production of inflammatory mediators, thereby reducing neutrophil infiltration. This point is now discussed in the revised manuscript lines 181-193.

The authors fail to establish a convincing link between the severity of the lesions and the reported neutrophil-cDC1-CD8+ T cell axis. The numbers of CD8+ T cells in the skin lesions appears to be unaffected by either lack of nociceptive neurons or neutrophil depletion and it remains unclear how the more severe lesions or the lack of tissue repair in the NAV1,8-DTA mice is caused.

- Our data support a model in which Nav1.8⁺ sensory neurons inhibit the production of inflammatory cytokines in the skin after HSV-1 infection. This reduction is associated with a down-regulation of neutrophil influx in the skin which is required to promote a robust CD8 T cell response. The T cell response is affected in the DLN and in the spleen but not in the skin (new Figure 3 and Suppl. Fig. 6F). Therefore we don't think that the increase in lesion size could be a consequence of the CD8 T cell defect. In Nav1.8-DTA mice, neutrophils were shown to be involved in both the defect in tissue repair and the defect in DC/CD8+T cell responses but these could be parallel phenomena without any causal link. To clarify this point, the two phenotypes are now described in two separated paragraphs in the manuscript. This point is also discussed lines 279-288 and 413-415 of the revised manuscript.

Our data suggest that the tissue lesions and the lack of tissue repair are due to the higher numbers of cells (monocytes and neutrophils) producing inflammatory mediators in nociceptor-deficient mice (new Figure 2). These data are consistent with the reduction in lesion size observed in Nav1.8-DTA mice upon neutrophil depletion (Fig. 4C).

The authors show accumulations of other immune cell subsets which could conceivably affect the course of HSV-1 infection and T cell priming / expansion as well. Monocyte numbers are decreased in mice lacking nociceptive neurons (Fig. 3B) as are cDC2 although the data is

contradictory (significant in Fig 3D, non-significant in 4C). This should at the very least be discussed and does weaken the causal link the authors report.

- The data presented in the initial Figure 3D showed some differences in the relative frequencies of the different cell types (expressed as the % of CD45⁺ cells) at day 8 pi. This phenotype was mainly the consequence of an increase in both the % and the absolute numbers of neutrophils. We now provide additional analysis on day 6 pi, which are consistent with this observation. We dissected the skin immune response in nociceptor-deficient and -sufficient mice on day 6 pi. This analysis revealed a large increase in the relative frequency of neutrophils in CD45⁺ skin cells of nociceptor-deficient mice (Figure 2B,C). Neutrophil counts were also increased in the skin of Nav1.8-DTA mice whereas the numbers of monocytes, macrophages, DC, $\gamma\delta$ T cells, eosinophils, mast cells were similar or only slightly modified compared to control mice (new Figures 2H, new Supplemental Figure 2C). Nav1.8⁺ sensory neurons are thus required to limit neutrophil infiltration in the skin in the context of HSV-1 infection.

Furthermore, the additional Figure 2 below show that neutrophil depletion, which restored the T cell response in Nav1.8-DTA mice did not affect the number of monocytes, macrophages or eosinophils, suggesting that these cell types do not play a major role in this regulation.

Additional Figure 2 (for reviewers only): Absolute cell numbers in the skin of HSV-1-infected control DTA (black bars) and Nav1.8-DTA (white bars) mice treated with isotype control antibody (left-side) or depleted of neutrophil (right-side) at the indicated time points after infection. Results are representative of three independent experiments. Each dot represents the data obtained for a single mouse. The results shown are representative of three independent experiments. The data shown are the mean \pm SEM.

Functional assessment of the immune populations in skin and lymph nodes might strengthen the hypothesis of the authors and provide an explanation for the increase inflammation and tissue damage beyond the numerical enrichment of immune cells.

- We now provide new functional data. We analyzed the cellular source of the inflammatory cytokines produced in excess in the skin of Nav1.8-DTA mice. Intracellular staining and flow cytometry analysis of skin cells on day 6 pi revealed increased production of TNF- α and IL-1 β by monocytes from Nav1.8-DTA mice (new Figures 2D, I). The percentages of inflammatory cytokine production by neutrophils were similar in DTA and Nav1.8-DTA mice, showing that sensory neurons are not required to regulate these functions (new Figure 2E). However, because the absolute number of neutrophils increased (new Figure 2 F), the numbers of TNF- α -, IL-6- and IL-1 β -producing neutrophils were higher in nociceptor-deficient mice than in their littermate controls (Figures 2G). By contrast, for the other immune cell subsets, the % and absolute numbers of cytokine-producing cells were similar in the two genotypes (data not shown and new supplemental Figures 3A-C). These data show that nociceptive sensory neurons down-regulate inflammatory cytokine production by monocytes and control neutrophil influx in the skin post-HSV-1 infection. These regulations are required to limit the cutaneous levels of TNF- α , IL-6 and IL-1 β pi.

However, we showed that neutrophil depletion was sufficient to reduce tissue damage in mice deficient in nociceptors, showing that although nociceptors can control monocyte activation, the main inflammatory phenotype requires the presence of neutrophils.

Minor points

Fig 2c,d,: skin lesions vary a lot, how are skin samples taken, processed and normalized for pfu, and for measurement of cytokines and chemokines? Information is lacking also in M&Ms

- It is true that the size of skin lesions can be variable between mice. As mentioned above, we increased the number of mice analyzed in new Figures 1B and Suppl. Figure 1D.

Regarding the measurement of cytokines/ chemokines and viral load, as mentioned in the method section, 1.5 cm x 2 cm skin rectangles surrounding the infection or scarification sites were analyzed. This is now described in more details (lines 541-545) in the revised manuscript.

Fig 3a/b: a tSNE figure is not necessarily helpful in this context; a clear figure showing gating strategy in combination with the quantification in 3c would be more helpful.

- The Figure 3 has been removed and replaced by a new Figure 2 which illustrates the increased neutrophil infiltrate in the Nav1.8-DTA mice on day 6 pi. The precise flow cytometry gating strategy is provided in new Supplemental Figure 2B.

Fig 4b: The more relevant comparison and visualization would be NAV1,8-DTA with or without neutrophil depletion in order to make a point about the role of neutrophils in inducing tissue damage.

- We performed additional analysis on lesion size (see our answers to questions 3 and 5, above) and as suggested by the reviewer we present the comparison of the lesion size, with or without neutrophil depletion, for each genotype. These data are shown in the new Figures 4C and D.

Fig4d: Viral titer in this setup should be shown to ensure that increased numbers of XCR1+ DC and OT-1 T cells are not due to the mice failing to control the infection due to their lack of neutrophils.

- Earlier studies (Wojtasiak et al., 2010) revealed that neutrophil depletion does not affect viral replication. Moreover, as discussed in the manuscript, Hor et al. demonstrated that neutrophil depletion in WT mice does not affect the CD8 T cells response to HSV-1 infection (Hor et al., 2017). Our data are consistent with these studies as neutrophil depletion did not affect the CD8 T cell response in DTA control mice (new Figure 4 F). Moreover, viral replication was similar in DTA and Nav1.8-DTA mice (Figure 1C). Therefore, even if the neutrophil response is exacerbated in Nav1.8-DTA mice it does not affect viral clearance. Altogether, these data do not favor a role of viral load differences to explain the restoration of DC and CD8 T cell responses in neutrophil-depleted Nav1.8-DTA mice. Therefore, in order to follow our Ethics Committee's recommendation to reduce the number of mice used in our study to the minimum required (the "Three Rs" rule), we did not conduct additional experiments on this point.

S1b: It is unclear what was chosen as a 'starting weight' since the DTA mice (presumably the 'control mice' which are mentioned as being used to determine 'starting weight') already start the experiment a 0.9 relative weight.

- We thank the reviewer for this remark. The figure is now corrected (New suppl. Fig. 1E).

S2d: Editing error in axis name

S1A “after scarification with PBS”, not “after infection”

- These points have been corrected in the revised manuscript (see new suppl. Fig. 1D)

Fig S4 (and others) statistics are lacking

- The DC response in skin DLN has been analyzed at earlier time points pi as suggested by the reviewers (new suppl. Fig. 5). This analysis was more relevant and the initial Fig .S4 has been removed.

S4B: FACS plots cannot be from same staining, CD24 levels/MFI are very different in the different plots

- We apologize for this error.

Reviewer #2 (Neuro-immune crosstalk)(Remarks to the Author):

This is an interesting study, which contains information regarding the effect/role of parenchymal cells of the nervous system in controlling HSV-1 processes. The authors demonstrated that the CD8 T-cell response to HSV-1 is impaired in nociceptor-deficient mice, which displayed enhanced skin inflammation upon HSV-1 infection. Their interpretation of the results as that the nociceptor is potentially a possible molecule involved in the interaction of T cells, which was supported by the observations that the nociceptor-deficient mice had an impaired CD8 T-cell response to HSV-1, and that lack of nociceptive sensory fibres in Nav1.8-DTA mice did not affect the viral clearance.

- We thank the reviewer for his/her positive feedback.

The paper would be more convincing if the following points can be addressed more clearly:

1. The study used a virus with a mutation of the TK gene that expressed the SIINFEKL-peptide derived from ovalbumin (OVA) and assessed the viral responses relied on responses of transferred OT-I T cells. It is not clear whether the intensity of OT-I responses completely

parallel with the real viral responses of HSV-1 infection and whether this augmented responses is antigen-specific, or it is restricted to viral responses only.

- This HSV-OVA model has been used in several recent manuscript from the lab of F. Carbone (co-author of this study) (Mackay et al. 2013; Park et al. 2018). This lab did experiments working out the optimum range for transgenic T cell number for adoptive transfer experiments in order to mimic the endogenous T cell response (Stock et al., 2007). They also found that naïve OT-I were transferred at a relatively low number (5×10^4) before infection, induces a T cell response which mimics the endogenous response to the viral antigen gB. In the present study, we used protocols based on these previous data.

2. It remained to be tested whether blockade of nociceptive effect could modulate intensity of HSV response. Such attempts should further support the predicted hypothesis.

We agree with the reviewer that blockade of nociceptive effect would reinforce our findings. Efficient blockade of sensory neurons activity could be achieved pharmacologically by injecting local anesthetics. However, as we are studying a process that takes place over a long period of time (8 days), repeated injections would be required to efficiently block nociceptive effects. Repeated injections into the skin could by itself induce inflammation, which could have important consequences on the local immune response and induce biases in the interpretation of the results, especially in a context where the tissue is already inflamed. Genetic ablation helps to avoid such problems. Indeed, the genetic model used in our study Nav1.8-DTA model has been previously used in numerous studies to demonstrate a role of nociceptive sensory neurons in various biological conditions (Abrahamsen et al., 2008; Chiu et al., 2013; Riol-Blanco et al., 2014; Talbot et al., 2015). We therefore believe that this genetic model, which is commonly used, is widely accepted as a model to study the role of sensory neurons expressing Nav1.8.

3. The authors analyzed the virus-specific T-cell response in the skin draining lymph nodes by monitoring the expansion of the transferred OT-I T cells, wondering whether how the responses in the skin reflects the immune responses taken place in the CNS.

- We did not address this question because we used a virus which only replicates in skin cells but not in neurons. We therefore focused on the immune response occurring in the skin and the skin DLN.

4. Mechanistic studies showed that cytokine/chemokine levels were higher in infected Nav1.8-

DTA mice. However, such changes may just reflected a stronger immune response. It is not examined whether producing cells also involved parenchymal cells in addition to infiltrating immune cells, and whether the increased cytokine production was caused by over-infiltration of PMN or reduced monocyte-infiltration, which may also cause greater immune responses, due to that an over infiltration of monocytes is also inhibitory.

- We thank the reviewer for this important question. We now provide additional data identifying the cellular source of inflammatory cytokines in mouse skin on D6 pi (New Figure 2, new Suppl. Figure 2). We analyzed the cellular source of the inflammatory cytokines produced in excess in the skin of Nav1.8-DTA mice. Intracellular staining and flow cytometry analysis of skin cells on day 6 pi revealed increased production of TNF- α and IL-1 β by monocytes from Nav1.8-DTA mice (new Figures 2D, I). The percentages of inflammatory cytokine production by neutrophils were similar in DTA and Nav1.8-DTA mice, suggesting that sensory neurons are not required to regulate their cytokine production (new Figure 2E). However, because the absolute number of neutrophils increased (new Figure 2F), the numbers of TNF- α -, IL-6- and IL-1 β -producing neutrophils were higher in nociceptor-deficient mice than in their littermate controls (Figures 2G). By contrast, for the other immune cell subsets, the % and absolute numbers of cytokine-producing cells were similar in the two genotypes (data not shown and new supplemental Figures 3A-C). Furthermore, we did not detect significant cytokine production in CD45-negative cells in these mice (data not shown). These data show that nociceptive sensory neurons down-regulate inflammatory cytokine production by monocytes and control neutrophil influx in the skin post-HSV-1 infection. These regulations are required to limit the cutaneous levels of TNF- α , IL-6 and IL-1 β pi.

Overall the paper is clearly presented. Demonstration of a correlation of nociceptive and viral intensity is interesting; however, the mechanisms leading to augmented immune responses could be more sophisticated.

- We thank the reviewer for this positive comment. We now provide additional functional data dissecting the functional mechanisms involved in new Fig. 2 and 3C,D.

Reviewer #3 (HSV, viral immunity)(Remarks to the Author):

Filtjens and colleagues used an established murine skin scarification protocol to investigate the contribution of nociceptive sensory neurons and neutrophils to mount robust CD8 T cell priming and robust adaptive immune response against herpes simplex virus type 1 (HSV). This

study provides further support to the emerging concept that the nervous system has important functions in addition to the immune system to control inflammation during an infection. The authors show that there was a defective primary CD8 T cell response against HSV leading to an exacerbated disease pathology associated with extensive skin lesions and an enhanced inflammatory cytokine and chemokine production in mice ablated for Nav1.8+ sensory neurons. Further experiments indicate that the neurons were required to downregulate the number of neutrophils to the infected skin, to control the skin XCR1 dendritic cell response, and to limit the tissue damage. Overall, the authors designed, controlled and executed their experiments well. However, important information is missing in some instances, which prevents the manuscript from being comprehensible to potential readers from the diverse fields of virology, neurobiology and immunology.

- We thank the reviewer for these positive comments.

Major Comments

1. Introduction, page 3 and 4: Could the authors please specify what kind of mediators are locally released by nociceptive sensory neurons upon activation that could modulate the activity of immune cells? Which mediators have proinflammatory and which ones anti-inflammatory activity?

- We clarified this point in the introduction (lines 53-54) and we referred to a very complete review. This point is also discussed in the discussion lines 444-447.

2. Introduction page 4 bottom, results page 5, discussion page 12/13, and/or Exp Procedures page 17: The authors need to provide a better introduction and characterization of the genetic mouse model used to ablate the nociceptor neurons. Neurons other than nociceptors express Nav1.8 in the mouse. Up to 75% of the neurons in dorsal-root ganglia express Nav1.8 Cre (Shields et al. 2012, Pain). The authors need to provide evidence, which neurons were lacking upon ablation in the skin and in the dorsal root ganglia; for example, by histology studies to characterize the neurons and neuronal processes that remain in the skin and DRG after ablation.

- The two graphs below (extracted from Usoskin et al., 2015) provide a comparison of the expression profiles of Nav1.8 (*Scn10a*) and Nav1.7 (*Scn9a*). As one can see Nav1.7 is expressed in all DRG neurons, whereas Nav1.8 expression is restricted to small diameter peptidergic and nonpeptidergic neurons and totally excluded from large neurons known to express Neurofilament heavy chain (NF200) (Usoskin et al., 2015).

[Redacted]

Moreover, in their Science paper, Abrahamsen et al. (2008) showed that developmental ablation of transient and persistent Nav1.8-expressing neuronal spared the vast majority of NF-200 expressing neurons. Given that we used the exact same ablation strategy as Abrahamsen et al, we expect that all peripherin⁺ neurons are ablated and the majority of NF200⁺ large neurons are spared. We clarified this point in the discussion of the revised manuscript (line 437-444).

Which skin cells in addition to the nociceptor neurons expressed Nav1.8+? Which immune cells expressed Nav1.8+? To what extent were such Nav1.8+ expressing cells ablated in this model?

- Microarray and RNA-Seq data available in public databases (<http://www.immgen.org>; <http://biogps.org>) show that the mRNA of *Scn10a* gene (encoding Nav1.8) is not expressed in immune cell subsets. To confirm this point we performed a fate mapping analysis by crossing the Nav1.8^{Cre} mice to Rosa26-TdTomato mice. In this model, the cells expressing Nav1.8 at one point of their life will express the fluorescent marker. Immunofluorescent analysis in the skin of these mice (new Supplemental Figure 1B) clearly showed that only nerve fibres but not other cell types express the fluorescent marker in the skin. These data were confirmed and completed by flow cytometry analysis showing that CD45⁺ hematopoietic cells in the skin, the LN and the spleen did not express the fluorescent marker (new Supplemental Figure 1C). These data confirmed the lack of expression of Nav1.8 in immune cells and reinforced the relevance of this genetic model.

What is the genetic background of the mice used in this study?

- The mouse genetic background is C57BL/6. This information has been added in the method section.

3. Results page 5, discussion page 12, Exp Procedures page 17: The authors need to provide a better description of the HSV-1 mutant that they have used in this study. Which HSV strain was used as parental, and what is the strain history of the HSV-TKminus mutant? Where in the genome or in which protein has the OVA SIINFEKL epitope been integrated? How was the inoculum prepared and purified? Cell extract, extracellular medium, gradient purification?

- This information has been added in the revised manuscript (lines 494-508).

4. Results page 7: Could the authors provide a rationale for their selection of cytokines that they have measured? What are the sources and the functions of IL-1beta, IL-6, TNFalpha, IFN-beta, GM-CSF, CXCL1, CCL3, or CCL2 in this infection model?

- HSV-1 infection induces the expression of inflammatory chemokines and cytokines, including CCL2, CCL3, CXCL1, IL-1 and IL-6, TNF- α , and type I IFNs (Paludan, 2001; Hukkanen et al., 2002; Melchjorsen et al., 2003; Finberg et al., 2005; Kawai and Akira, 2006; Wuest and Carr 2008; Rosato and Leib, 2015)

To address the reviewer question, we performed additional experiments and analyzed the cellular source of the main inflammatory cytokines IL-1 β , IL-6 and TNF α , which are known to act on nociceptor neurons to sensitize pain pathways (Baral et al., 2019). We performed intracellular staining and flow cytometry analysis on skin cells from DTA and Nav1.8-DTA 6 days post-infection. These new data are shown in new Fig. 2 and new Suppl. Fig. 3.

6. Results pages 7ff, Figure 2 and Figure 3: There were similar numbers of neutrophils in the skin at 4 dpi in Nav1.8+ and Nav1.8-ablated mice, while there were more neutrophils in the Nav1.8-ablated mice at 8 dpi. The authors concluded that more neutrophils had been recruited to the skin in the Nav1.8 ablated mice. However, an alternative hypothesis is that more neutrophils had been unable to leave the skin. Several groups have reported the phenomenon of neutrophil reverse migration, and suggested that the expression of hypoxia inducible factor 1 alpha might contribute to an increased retention of neutrophils, which would also delay the resolution of inflammation.

The authors could investigate the level of HIF-1alpha in their system to address whether it contributes to neutrophil behaviour and inflammation. Furthermore, the high levels of cytokine expression in the skin in the absence of Nav1.8 may contribute more than the increased neutrophil numbers to the delayed resolution of inflammation.

- These points are well taken and are now discussed in the revised manuscript (lines 264-277). We further analyzed the mechanisms by which neutrophil infiltration persisted in Nav1.8-DTA mice on day 6 pi (new Fig. 2, 4A). A higher number of neutrophils could be the results of an increase in their recruitment, a defect in cell death or a higher level of retention in the tissue.

As suggested by the reviewer, we have measured HIF-alpha in the skin of Nav1.8-DTA and control mice on day 6 pi. We did not detect any difference in the level of HIF-1alpha in the skin of Nav1.8-DTA and control DTA mice (new Fig.S6C). Consistently, the frequency of neutrophil undergoing cell death in the skin of these mice was also similar in the two genotypes (new Fig. S6D). These data suggest that the recruitment and not the survival of neutrophils is increased in the absence of nociceptive sensory neurons, which is in agreement with the higher level of CXCL1 in the skin of Nav1.8-DTA mice (Figure 1D).

We also show in the new Figure 2 that increased neutrophil numbers contribute to the high levels of cytokine expression in the skin in the absence of Nav1.8⁺ neurons. These two phenomena are therefore closely linked.

8. The authors show the effects that ablating Nav1.8⁺ neurons have on cytokine expression, neutrophil levels in the skin, and CD8 T cell response against HSV. It would be interesting to know how activating Nav1.8 with agonists or with optogenetic methods would influence the immune system in this system.

- We agree with the reviewer that these are interesting questions but we think they are beyond the scope of the present study.

9. Pro-inflammatory cytokines, like TNF, can sensitize sodium channels including Nav1.8. The authors need to discuss whether the high level of TNF in the Nav1.8-ablated mice could be the consequence of this lack of Nav1.8 sensitization.

- As mentioned by the reviewer, cytokines can directly act on nociceptor neurons to sensitize pain pathways (Baral et al., 2019). Pro-inflammatory cytokines, such TNF- α and IL-1 β , can be pro-algesic by directly driving nociceptor activation (Cook et al., 2018). In particular, TNF- α participates to the sensitization of voltage gated sodium channels (VGSCs), including the TTX resistant Nav1.8. Tissue injury, inflammation or infection can therefore lead to the activation of the immune system which then leads to the activation and hypersensitization of sensory neurons. The reviewer proposed that high TNF- α levels could, in turn, be explained by the lack of Nav1.8 sensitization in Nav1.8-DTA mice. We totally agree with the reviewer. We now clearly show that Nav1.8⁺ neurons are required to down regulate TNF- α and IL-1 β production by

monocytes (new Figure 2D). This point is further discussed in the revised version of the manuscript.

Minor Comments

1. Discussion, page 12: Last sentence is unclear.

- We apologize for this error. This sentence has been removed.

5. Exp procedures page 17: How were the flank scarification and HSV infection performed? Amount of virus, time of inoculation, and duration of infection?

- We provided more details on these procedures in the method section (lines 508-519).

References

Abrahamsen, B., Zhao, J., Asante, C.O., Cendan, C.M., Marsh, S., Martinez-Barbera, J.P., Nassar, M.A., Dickenson, A.H., and Wood, J.N. (2008). The cell and molecular basis of mechanical, cold, and inflammatory pain. *Science*. 321, 702–705.

Allan, R.S., Waithman, J., Bedoui, S., Jones, C.M., Villadangos, J.A., Zhan, Y., Lew, A.M., Shortman, K., Heath, W.R., and Carbone, F.R. (2006). Migratory Dendritic Cells Transfer Antigen to a Lymph Node-Resident Dendritic Cell Population for Efficient CTL Priming. *Immunity* 25, 153–162.

Appelberg, R. (2007). Neutrophils and intracellular pathogens: beyond phagocytosis and killing. *Trends Microbiol.* 15, 87–92.

Baral, P., Udit, S., and Chiu, I.M. (2019). Pain and immunity: implications for host defence. *Nat. Rev. Immunol.* 19, 433–447.

Bedoui, S., Whitney, P., Waithman, J., Eidsmo, L., Wakim, L., Caminschi, I., Allan, R.S., Wojtasiak, M., Shortman, K., Carbone, F.R., et al. (2009). Cross-presentation of viral and self antigens by skin-derived CD103+ dendritic cells. *Nat. Immunol.* 10, 488–495.

Chiu, I.M., von Hehn, C.A., and Woolf, C.J. (2012). Neurogenic Inflammation – The Peripheral Nervous System’s Role in Host Defense and Immunopathology. *Nat. Neurosci.* 15, 1063–1067.

Chiu I.M., Heesters B.A., Ghasemlou N., von Hehn C.A., Zhao D., Tran J., Wainger B., Strominger A., Muralidharan S., Horswill A.R., Wardenburg J.B., Hwang S.W., Carroll M.C.

- and Woolf C.J. (2013). Bacteria activate sensory neurons that modulate pain and inflammation. *Nature*. 52-57
- Cook, A.D., Christensen, A.D., Tewari, D., McMahon, S.B., and Hamilton, J.A. (2018). Immune Cytokines and Their Receptors in Inflammatory Pain. *Trends Immunol.* 39, 240–255.
- Eidsmo, L., Allan, R., Caminschi, I., van Rooijen, N., Heath, W.R., and Carbone, F.R. (2009). Differential Migration of Epidermal and Dermal Dendritic Cells during Skin Infection. *J. Immunol.* 182, 3165–3172.
- Finberg, R.W., Knipe, D.M., and Kurt-jones, E.A. (2005). Review Herpes Simplex Virus and Toll-Like Receptors. *Viral Immunol.* 18, 457–465.
- Hogquist, K.A., Jameson, S.C., Heath, W.R., Howard, J.L., Bevan, M.J., and Carbone, F.R. (1994). T Cell Receptor Antagonist Peptides Induce Positive Selection. *Cell* 76, 17–27.
- Hor, J.L., Whitney, P.G., Brooks, A.G., William, R., and Mueller, S.N. (2015). Spatiotemporally Distinct Interactions with Dendritic Cell Subsets Facilitates CD4 + and CD8 + T Cell Activation to Localized Viral Infection. *Immunity* 43, 554–565.
- Hor, J.L., Heath, W.R., and Mueller, S.N. (2017). Neutrophils are dispensable in the modulation of T cell immunity against cutaneous HSV-1 infection. *Sci. Rep.* 7, 1–13.
- Hukkanen, V., Broberg, E., Salmi, A., and Erälina, J.P. (2002). Cytokines in experimental herpes simplex virus infection. *Int. Rev. Immunol.* 21, 355–371.
- Kashem SW, Riedl MS, Kaplan DH (2015). Nociceptive sensory fibers drive interleukin-23 production from CD301b+ dermal dendritic cells and drive protective cutaneous immunity. *Immunity* 43, 515–526.
- Kawai, T., and Akira, S. (2006). Innate immune recognition of viral infection. *Nat. Immunol.* 7, 131–137.
- Mackay LK, Rahimpour A, Ma JZ, Collins N, Stock AT, Hafon ML, Vega-Ramos J, Lauzurica P, Mueller SN, Stefanovic T, Tschärke DC, Heath WR, Inouye M, Carbone FR, Gebhardt T. The developmental pathway for CD103(+)CD8+ tissue-resident memory T cells of skin. *Nat Immunol.* 2013 Dec;14(12):1294-301. doi: 10.1038/ni.2744.
- Melchjorsen, J., Sørensen, L.N., and Paludan, S.R. (2003). Expression and function of chemokines during viral infections: from molecular mechanisms to in vivo function. *J. Leukoc. Biol.* 74, 331–343.
- Odobasic, D., Kitching, A.R., Yang, Y., O’Sullivan, K.M., Muljadi, R.C.M., Edgton, K.L., Tan, D.S.Y., Summers, S.A., Morand, E.F., and Holdsworth, S.R. (2013). Neutrophil

myeloperoxidase regulates T-cell-driven tissue inflammation in mice by inhibiting dendritic cell function. *Blood* 121, 4195–4204.

Paludan, S.R. (2001). Requirements for the Induction of Interleukin-6 by Herpes Simplex Virus-Infected Leukocytes. *J. Virol.* 75, 8008–8015.

Park SL, Zaid A, Hor JL, Christo SN, Prier JE, Davies B, Alexandre YO, Gregory JL, Russell TA, Gebhardt T, Carbone FR, Tschärke DC, Heath WR, Mueller SN, Mackay LK. Local proliferation maintains a stable pool of tissue-resident memory T cells after antiviral recall responses. *Nat Immunol.* 2018 Feb;19(2):183-191. doi: 10.1038/s41590-017-0027-5.

Riol-Blanco, L., Ordoñas-Montanes, J., Perro, M., Naval, E., Thiriot, A., Alvarez, D., Paust, S., Wood, J.N., and Von Andrian, U.H. (2014). Nociceptive sensory neurons drive interleukin-23-mediated psoriasiform skin inflammation. *Nature* 510, 157–161.

Rosato, P.C., and Leib, D.A. (2015). Neuronal Interferon Signaling Is Required for Protection against Herpes Simplex Virus Replication and Pathogenesis. *PLoS Pathog.* 11, 1–22.

Soehnlein, O., Weber, C., and Lindbom, L. (2009). Neutrophil granule proteins tune monocytic cell function. *Trends Immunol.* 30, 538–546.

Stock, A.T., Mueller, S.N., Lint, A.L. Van, Heath, W.R., and Carbone, F.R. (2004). Cutting Edge : Prolonged Antigen Presentation after. *J. Immunol.* 173, 2241–2244.

Stock, A.T., Mueller, S.N., Kleinert, L.M., Heath, W.R., Carbone, F.R., and Jones, C.M. (2007). Optimization of TCR transgenic T cells for in vivo tracking of immune responses. *Immunol. Cell Biol.* 85, 394–396.

Talbot S., Abdulnour R.E., Burkett P.R., Lee S., Cronin S.J.F., Pascal M.A, Laedermann C., Foster S.L., Tran J.V., Lai N., Chiu I.M., Ghasemlou N., DiBiase M., Roberson D., von Hehn C., Agac B., Haworth O., Seki H., Penninger J.M., Kuchroo V.K., Bean B.P., Levy B.D. and Woolf C.J. (2015) Silencing nociceptor neurons reduces allergic airway inflammation. *Neuron.* 82, 341-354

Tenser, R.B. (1991). Role of Herpes simplex Virus Thymidine Kinase Expression in Viral Pathogenesis and Latency. *Intervirology* 32.

Usoskin, D., Furlan, A., Islam, S., Abdo, H., Lönnberg, P., Lou, D., Hjerling-leffler, J., Haeggström, J., Kharchenko, O., Kharchenko, P. V, et al. (2015). Unbiased classification of sensory neuron types by large-scale single-cell RNA sequencing. *Nat. Neurosci.* 18, 145–153.

Van Lint, A., Ayers, M., Brooks, A.G., Coles, R.M., Heath, W.R., and Carbone, F.R. (2004).

Herpes Simplex Virus-Specific CD8+ T Cells Can Clear Established Lytic Infections from Skin and Nerves and Can Partially Limit the Early Spread of Virus after Cutaneous Inoculation. *J. Immunol.* 172, 392–397.

Wojtasiak, M., Pickett, D.L., Tate, M.D., Bedoui, S., Job, E.R., Whitney, P.G., Brooks, A.G., and Reading, P.C. (2010). Gr-1+ cells, but not neutrophils, limit virus replication and lesion development following flank infection of mice with herpes simplex virus type-1. *Virology* 407, 143–151.

Wuest, T.R. and D. J.J Carr (2008). The role of chemokines during herpes simplex virus-1 infection. *Front. Biosci.* 13, 4862-72.

REVIEWER COMMENTS

Reviewer #1 (Remarks to the Author):

In their revised manuscript, Filtjens et al. emphasize context-dependent role for nociceptive neurons in controlling inflammation of the skin in their viral infection model. The points and questions raised were generally addressed convincingly and several pieces of key data have been added, clarifying the kinetics of the different processes and the possible influence of other immune cells.

There are nevertheless still a few points and inconsistencies that should be addressed:

- Title: the new title expands findings to “antiviral T cell responses” in general, although only the response to HSV has been tested (CFA-OVA is not a virus). It remains unclear if the regulation of local inflammation versus the regulation of T cell priming and expansion in LN and spleen (but not in the skin!) is mechanistically linked or represents two independent processes. I would therefore highly recommend to stick to the original, more appropriate title.

- New OVA-CFA model Fig 4: as of M&Ms, CFA is injected into the footpad, which is usually considered subcutaneous? Is this really a skin inflammation model? Authors should discuss and clearly state also in results where and how it is injected.

- “We decided not to use a model in which Nav1.8 neurons express the diphtheria toxin receptor (iDTR) because this model requires the injection of a toxin that will induce neuronal cell death in the tissue, potentially inducing inflammatory reactions that could interfere with the processes we are studying.” Authors should then at least discuss the potential limitation that no bona fide conditional model has been used.

- “did not detect any activation of the IL-23/IL-17 axis in these mice (this point is now stated in the paper lines 153...” – as the IL23/IL17 axis is much studied in the context of nociceptive neurons, the Authors should consider showing these data in supplements.

- I would further suggest including additional Figure 2 which was presented in the rebuttal in the supplementary of the paper given that the effect of neutrophil depletion on other immune cell population is of major relevance.

- Authors have changed Figures that “had errors” and have pooled several experiments that were different when different investigators did the experiments. For example new Fig 1b and 4C: corresponding groups have very different lesion sizes and kinetics; it is also very different from the original Fig 4B, where lesions in NAV-DTA mice were quite significant on d2. Authors should state very clearly how many data points from how many mice and individual repeat experiments have been pooled, and, as figures representing key findings obviously have a large variability depending on the investigator, authors should show all individual experiments showing lesion sizes in the supplements.

There are additional inconsistencies in the Figures:

- Figure 2e Neutrophil TNF α ; the replicate used to show TNF α in the FACS plots (9.12%) does not appear in the scatter plot to the right – how can this be? In addition, the MFI in the NAV1,8-DTA Neutrophils appears strongly increased compared to the WT and with slightly different gating there might be a significant difference between the genotypes.

- Moreover, the CD45 MFI in NAV-DTA mice is diminished in the TNF and IL6 sample, but not IL1b

sample, are these really from the same experiments?

- Figure 2a: Are TCRab+ CD4+ T cells enriched in the NAV1,8-DTR mice? The data in the tSNE plot implies they might be. They are the only subset on the tSNE plot which is not quantified on a bar graph and I would suggest to add this data for the sake of completeness. An altered CD4 response might also add context to the changes in CD8 responses that were observed.

Reviewer #2 (Remarks to the Author):

The revised manuscript is significantly improved. The authors' responses to the previously raised concerns are reasonable. This reviewer did not have further questions.

Reviewer #3 (Remarks to the Author):

In my assessment, Filtjens, Roger et al. have addressed all reviewers' responses in their revised manuscript, despite the challenging pandemic times. They have included additional experiments or have added further references from the literature to explain the text revisions made.

Point by point response to the reviewers' comments (Second stage of review)

We thank the editors and reviewers for their enthusiasm about our study. We appreciated the last points raised by the reviewer #1, which allowed us to improve our study even further. We have modified our manuscript to address the points raised.

The additional changes to the text, for this second phase of revision, are presented in red in the revised manuscript. Please find below (in red) a point-by-point response to the last reviewer's concerns, explaining how they have been addressed in the revised manuscript.

REVIEWERS' COMMENTS

Reviewer #1 (Remarks to the Author):

In their revised manuscript, Filtjens et al. emphasize context-dependent role for nociceptive neurons in controlling inflammation of the skin in their viral infection model. The points and questions raised were generally addressed convincingly and several pieces of key data have been added, clarifying the kinetics of the different processes and the possible influence of other immune cells.

- We thank the reviewer for this positive feedback on our revised manuscript.

There are nevertheless still a few points and inconsistencies that should be addressed:

- Title: the new title expands findings to "antiviral T cell responses" in general, although only the response to HSV has been tested (CFA-OVA is not a virus). It remains unclear if the regulation of local inflammation versus the regulation of T cell priming and expansion in LN and spleen (but not in the skin!) is mechanistically linked or represents two independent processes. I would therefore highly recommend to stick to the original, more appropriate title.

- The restoration of the T cell response upon neutrophil depletion in Nav1.8-DTA mice strongly support a model in which the effect on T cell priming is linked to the regulation of inflammation (Figure 4). However, we changed the title as requested by the reviewer.

- New OVA-CFA model Fig 4: as of M&Ms, CFA is injected into the footpad, which is usually considered subcutaneous? Is this really a skin inflammation model? Authors should discuss and clearly state also in results where and how it is injected.

- Previous studies have shown that intraplantar injection of CFA induces skin inflammation (Ghasemlou et al., 2015; Caceres et al., 2017). These studies are consistent with our data showing an influx of neutrophils in the skin upon CFA intraplantar injection (Figure 4G). This point has been clarified in the results and more details have been added in the method section.

- “We decided not to use a model in which Nav1.8 neurons express the diphtheria toxin receptor (iDTR) because this model requires the injection of a toxin that will induce neuronal cell death in the tissue, potentially inducing inflammatory reactions that could interfere with the processes we are studying.”

fAuthors should then at least discuss the potential limitation that no bona fide conditional model has been used.

- We think that our genetic model of nociceptive neuron ablation is relevant to address the role of these neurons during HSV-1 infection. This Nav1.8-DTA model has been used in several other studies dissecting the role of these neurons in various pathological models (Abrahamsen et al., 2008; Chiu et al., 2013; Riol-Blanco et al., 2014; Talbot et al., 2015). However, we included a sentence in the revised manuscript (line 454-455), mentioning that complementary studies in other genetic models will be interesting in the future.

- “did not detect any activation of the IL-23/IL-17 axis in these mice (this point is now stated in the paper lines 153...” – as the IL23/IL17 axis is much studied in the context of nociceptive neurons, the Authors should consider showing these data in supplements.

- The skin levels of IL-17 and IL-23 were below the detection threshold of CBA assays that we used to measure the cytokine concentration (below 10 pg/ml). Therefore, it is difficult to show a figure as we cannot measure any concentration precisely. This point has been clarified in the text (lines 156-157) and we added a section in the methods to describe precisely the experimental conditions used to measure IL-17 and IL-23 production in the skin (lines 559-561).

- I would further suggest including additional Figure 2 which was presented in the rebuttal in the supplementary of the paper given that the effect of neutrophil depletion on other immune cell population is of major relevance.

- The data presented initially only in the additional Figure 2 of the rebuttal have been added in the new Supplemental Figure 6F and are now described lines 308-309 of the revised manuscript.

- Authors have changed Figures that “had errors” and have pooled several experiments that were different when different investigators did the experiments. For example new Fig 1b and 4C: corresponding groups have very different lesion sizes and kinetics; it is also very different from the original Fig 4B, where lesions in NAV-DTA mice were quite significant on d2. Authors should state very clearly how many data points from how many mice and individual repeat experiments have been pooled, and, as figures representing key findings obviously have a large variability depending on the investigator, authors should show all individual experiments showing lesion sizes in the supplements.

- The data presented in Figure 1B represent a pool of data obtained in 7 independent experiments performed by two different investigators. The lesion size can be quite variable between mice and between experimenters, most likely because the protocol includes a scarification step. Indeed, the pressure applied for scarification using a mechanical device may vary from experimenter to experimenter. However, the phenotype of Nav1.8-DTA mice

is very robust as it was observed in series of experiments performed by two independent investigators. We now provide a table (new Supplemental Table 1) in which all the raw data obtained for every individual mouse is shown. The corresponding experiments and the investigator who has performed them (#1 or #2) are also specified in the table. The total number of mice used in each experimental condition is specified in the figure legends.

- As you can see from the representative images shown in Figure 1A, the lesions can sometimes be heterogeneous and non-uniform in appearance. This is why it was essential that all photographs of lesions were re-analysed by a single person using exactly the same measurement criteria. This was a prerequisite for the results to be comparable. In the revised manuscript, all the analysis of lesion size were therefore done by the same investigator (#2) using photographs. The investigator was blinded to the genotype of each mouse during the analysis. This point is now clarified in the method section lines 529-532 and in the legend of the Supplemental Table 1.

- All the experiments presented in Fig. 4C were performed by a single investigator (#1) and the lesion size was reanalyzed by investigator #2 (as explained above). The new Figure 4C shows the raw data obtained for every individual mouse as requested by the reviewer.

There are additional inconsistencies in the Figures:

- Figure 2e Neurophil TNFa; the replicate used to show TNFa in the FACS plots (9.12%) does not appear in the scatter plot to the right – how can this be? In addition, the MFI in the NAV1,8-DTA Neutrophils appears strongly increased compared to the WT and with slightly different gating there might be a significant difference between the genotypes.

- In Figure 2E (right panels), the gating used to do the analysis of the frequency of TNF- α ⁺ cells was initially done on histogram plots. During the course of revision of the figures, we decided that it was clearer to show dot plot representations which visually highlight better the positive and negative populations. When these dot plots were generated, the gate was not placed exactly at the same position than in the histogram analysis, which generated slightly different percentages compared to the initial analysis (shown in the scatter plots to the right). This has now been corrected. We thank the reviewer for highlighting this point.

- We analyzed the fluorescence intensity of TNF- α staining in neutrophils from Nav1.8-DTA vs DTA mice (see Additional Figure 1 below). We did not detect any difference between the two genotypes.

Additional Figure 1 for reviewer only: This figure is related to the data presented in Figure 2E of the revised manuscript. *TNF- α* expression (GeoMean) in total neutrophil population (left panel) or in the *TNF- α* ⁺ gate (right panel).

- Moreover, the CD45 MFI in NAV-DTA mice is diminished in the TNF and IL6 sample, but not IL1b sample, are these really from the same experiments?

- Three independent experiments were done. The dot plots presented were from the same experiment. It is true that we observed some variability in the MFI of the CD45 staining between samples. However as highlighted in the graph below, we did not detect any significant difference between the two groups of mice (Additional Figure 2 below). In order that the representative figures illustrate more accurately this lack of difference, we have chosen to show samples for which the MFI of the CD45 staining was similar between the 2 groups of mice (see the revised version of the Figure 2E).

Additional Figure 2 for reviewer only: This figure is related to the data presented in Figure 2E of the revised manuscript.

- Figure 2a: Are TCRab⁺ CD4⁺ T cells enriched in the NAV1,8-DTR mice? The data in the tSNE plot implies they might be. They are the only subset on the tSNE plot which is not quantified on a bar graph and I would suggest to add this data for the sake of completeness. An altered CD4 response might also add context to the changes in CD8 responses that were observed.

- We added the results obtained on CD4 T cells in the new Supplemental Figure 2C. We observed no significant difference between the two genotypes. As for the other cell subsets, the differences in the t-SNE plots are due to the increase in the neutrophil absolute numbers (and thus percentages) in Nav1.8-DTA mice.

Reviewer #2 (Remarks to the Author):

The revised manuscript is significantly improved. The authors' responses to the previously raised concerns are reasonable. This reviewer did not have further questions.

- We thank the reviewer for his/her interest in our study as well as for his/her previous comments and suggestions which allowed us to considerably improve our manuscript.

Reviewer #3 (Remarks to the Author):

In my assessment, Filtjens, Roger et al. have addressed all reviewers' responses in their revised manuscript, despite the challenging pandemic times. They have included additional experiments or have added further references from the literature to explain the text revisions made.

- We thank the reviewer for his/her interest in our study as well as for his/her previous comments and suggestions which allowed us to considerably improve our manuscript.

References:

Abrahamsen, B., Zhao, J., Asante, C.O., Cendan, C.M., Marsh, S., Martinez-Barbera, J.P., Nassar, M.A., Dickenson, A.H., and Wood, J.N. (2008). The cell and molecular basis of mechanical, cold, and inflammatory pain. *Science* (80). 321, 702–705.

Caceres et al. (2017). Transient Receptor Potential Cation Channel Subfamily M Member 8 channels mediate the anti-inflammatory effects of eucalyptol. *British Journal of Pharmacology* 174 867–879,

Chiu I.M., Heesters B.A., Ghasemlou N., von Hehn C.A., Zhao D., Tran J., Wainger B., Strominger A., Muralidharan S., Horswill A.R., Wardenburg J.B., Hwang S.W., Carroll M.C. and Woolf C.J. (2013). Bacteria activate sensory neurons that modulate pain and inflammation. *Nature*. 52-57

Ghasemlou, N., Chiu, I.M., Julien, J.P., and Woolf, C.J. (2015). CD11b+Ly6G- myeloid cells mediate mechanical inflammatory pain hypersensitivity. *Proc. Natl. Acad. Sci. U. S. A.* 112, E6808–E6817.

Riol-Blanco, L., Ordovas-Montanes, J., Perro, M., Naval, E., Thiriout, A., Alvarez, D., Paust, S., Wood, J.N., and Von Andrian, U.H. (2014). Nociceptive sensory neurons drive interleukin-23-

mediated psoriasiform skin inflammation. *Nature* 510, 157–161.

Talbot S., Abdulnour R.E., Burkett P.R., Lee S., Cronin S.J.F., Pascal M.A, Laedermann C., Foster S.L., Tran J.V., Lai N., Chiu I.M., Ghasemlou N., DiBiase M., Roberson D., von Hehn C., Agac B., Haworth O., Seki H., Penninger J.M., Kuchroo V.K., Bean B.P., Levy B.D. and Woolf C.J. (2015) Silencing nociceptor neurons reduces allergic airway inflammation. *Neuron*. 82, 341-354